# Euclean: Automated Geometry Problem Formalization with Unified Verification in Lean

**Linbin Tang** [†12] **Jingyan You** [†12] **Zilin Kang** [†12] **Hanzhang Liu** [†32] **Sophia Zhang** [4] **Zenan Li** [5]
**Chenrui Cao** [†62] **Liangcheng Song** [†62] **Jiaao Wu** [†12] **Xian Zhang** [2] **Fan Yang** [2]

## Abstract

Recent formal reasoning systems have reached IMO-level performance, yet they leave a fragmented landscape: algebra and number theory are handled in Lean, while geometry still relies on domain-specific languages with limited formal guarantees. This split increases the trusted computing base and hinders unified model development. Existing geometry-in-Lean efforts (LeanEuclid, LeanGeo) introduce custom axiom systems incompatible with standard MATHLIB, and their small scale ($< 1{,}100$ problems) limits large-scale training. Native MATHLIB autoformalization of geometry, however, poses distinct challenges: implicit diagrammatic assumptions (e.g., topological configuration and non-degeneracy) must be made explicit rather than deferred to external solvers, and models must adapt to MATHLIB's small, rapidly evolving geometry infrastructure. We present EUCLEAN, a four-stage framework—constraint explication, configuration anchoring, formalization mapping, and iterative repair—for automatically formalizing geometry in native MATHLIB. We construct OMNI-Geometry (768 competition problems) and Numina-Geometry (177,597 problems), the largest geometry formalization dataset in Lean. Human evaluation shows 48.89% TOP1 and 73.33% TOP5 accuracy. Training Goedel v2 on our formalizations improves proof success from 13.6% to 15.1%, validating dataset quality for unified neural theorem proving. Code and datasets: https://github.com/tlb-22/Euclean.

## 1. Introduction

Formal reasoning has emerged as a cornerstone of reliable AI systems for mathematics, offering machine-verifiable guarantees that complement the impressive but sometimes unreliable outputs of large language models. Recent years have witnessed remarkable achievements: AlphaProof (Google DeepMind, 2024) and SeedProver (ByteDance Seed Team, 2025) have demonstrated IMO gold-medal level performance on algebra and number theory problems through formal theorem proving in Lean, while AlphaGeometry (Trinh et al., 2024) and SeedGeometry (ByteDance Seed Team, 2025) have achieved comparable results on geometry problems through domain-specific symbolic reasoning. These systems represented a major milestone at IMO 2024-2025, collectively solving problems at the level of human gold medalists.

However, this success reveals a fundamental fragmentation. Systems like AlphaProof and SeedProver operate within LEAN 4 and MATHLIB (de Moura & Ullrich, 2021; The mathlib Community, 2020), benefiting from a well-established proof assistant. In contrast, geometry systems employ custom DSLs lacking formal foundations. This divide increases the trusted computing base—geometry proof correctness depends on custom implementations that lack the scrutiny of mature proof assistants—and hinders unified model development by requiring separate training for geometry versus other domains.

The community has pursued geometry formalization in Lean. LeanEuclid (Murphy et al., 2024) and LeanGeo (Song et al., 2025) introduce synthetic axiom systems with SMT-based automation; MATP-BENCH (He et al., 2025) provides multimodal problems. However, these efforts introduce custom axioms isolated from standard MATHLIB's algebraic foundations, and their scale ($<1.1$K problems) is far below what enabled breakthroughs in other domains, such as LeanWorkbook (Ying et al., 2024) which contains 57K problems (83K in Plus version).

Native MATHLIB autoformalization poses unique challenges. Unlike existing custom systems that defer validity checks to external solvers, MATHLIB enforces strict preconditions

---

†Work done as an intern at Microsoft Research. [1]Tsinghua University, Beijing, China [2]Microsoft Research [3]New York University, New York, USA [4]Massachusetts Institute of Technology, Cambridge, USA [5]ETH Zürich, Zürich, Switzerland [6]University of Science and Technology of China, Hefei, China. Correspondence to: Xian Zhang <zhxian@microsoft.com>.

*Proceedings of the 43rd International Conference on Machine Learning*, Seoul, South Korea. PMLR 306, 2026. Copyright 2026 by the author(s).

as mandatory compilation guards. This turns *implicit diagrammatic assumptions*—such as topological configuration, relative positioning, and non-degeneracy—into rigid syntax barriers that must be explicitly resolved. Furthermore, the model must adapt to MATHLIB's geometric constructs that differ from informal notation, while navigating a geometry infrastructure that is smaller and evolves more rapidly than its algebraic counterparts, making traditional pretraining approaches difficult.

We present EUCLEAN, a framework addressing these challenges through: (1) a **four-stage formalization pipeline**—constraint explication, configuration anchoring, formalization mapping, and iterative repair—using `Plane := EuclideanSpace ℝ (Fin 2)` with "prove-first" reasoning for non-degeneracy conditions; and (2) **large-scale datasets**—OMNI-Geometry (768 competition problems) and Numina-Geometry (177,597 problems), the largest Lean geometry formalization to date.

We validate the quality of our formalized datasets through human expert annotation and downstream training experiments. Human evaluation on 180 randomly sampled OMNI-Geometry problems shows 48.89% TOP1 accuracy and 73.33% TOP5 accuracy. Crucially, our MATHLIB-native formalization enables immediate knowledge transfer: the Goedel v2 model (Lin et al., 2025), trained on general MATHLIB problems, achieves 13.6% proof success on Numina-Geometry without any geometry-specific training—demonstrating that unified MATHLIB compatibility allows existing theorem provers to generalize to geometry. Further training on our dataset improves this to 15.1%, validating its utility for advancing neural theorem proving.

In summary, our contributions are: (1) a four-stage geometry formalization pipeline that handles non-degeneracy conditions and maps to native MATHLIB constructs; (2) two large-scale datasets—OMNI-Geometry (768 problems) and Numina-Geometry (177,597 problems)—the largest Lean geometry formalization to date; and (3) extensive evaluation demonstrating formalization quality and downstream utility for neural theorem proving.

## 2. Related Work

**Automated Geometry Reasoning.** Automated reasoning in geometry spans algebraic methods (Wu, 1978), coordinate-based approaches (Chou, 1988), and synthetic methods (Gelernter, 1959). Recent neural approaches include AlphaGeometry (Trinh et al., 2024), which combines a language model with a symbolic deduction engine to achieve IMO-level performance,[1] TongGeometry (Zhang et al., 2024) with Monte Carlo tree search, and SeedGeome-

---

[1] AlphaGeometry's DSL has documented soundness issues (Lean Prover Community, 2025). See Appendix E for details.

try (ByteDance Seed Team, 2025). However, these systems operate within custom DSLs lacking the formal guarantees of established proof assistants.

**Neural Theorem Proving and Large-Scale Datasets.** The rapid progress in neural theorem proving is driven not only by algorithmic advances but critically by the emergence of large-scale formalization datasets. LEAN 4 (de Moura & Ullrich, 2021) with MATHLIB (The mathlib Community, 2020) has become the dominant platform, with its ecosystem growing rapidly in algebra and number theory. LeanDojo (Yang et al., 2023) provides retrieval-augmented proving environments; LeanWorkbook (Ying et al., 2024) contributes massive training corpora enabling models like DeepSeek-Prover-V2 (Ren et al., 2025) (88.9% on MiniF2F), Kimina-Prover (Numina & Kimi Team, 2025), and Goedel-Prover (Lin et al., 2025) to achieve state-of-the-art results. These systems target proof search given formal statements; our work addresses the upstream formalization problem for geometry.

**Geometry Formalization in Lean.** Concurrent work has explored geometry formalization in Lean. LeanEuclid (Murphy et al., 2024) introduced a benchmark based on System E (Avigad et al., 2009) with SMT-based proof automation. LeanGeo (Song et al., 2025) extends this with a larger theorem library. MATP-BENCH (He et al., 2025) provides 1,056 multimodal problems. However, LeanEuclid and LeanGeo operate outside standard MATHLIB due to custom axiom systems, limiting ecosystem integration (see Appendix F for detailed comparisons). More critically, these datasets (<1.1K problems) are far smaller than what enabled breakthroughs in algebra—LeanWorkbook contains 57K problems (83K in Plus version). Our Numina-Geometry (177,597 problems) aims to close this gap.

## 3. Methodology

We present EUCLEAN, a systematic framework for translating informal natural language geometry problems into rigorous, verifiable specifications in LEAN 4. Formally, given an informal natural language problem statement $\mathcal{P}$, our goal is to synthesize a corresponding formal Lean theorem $\mathcal{T}$ that is syntactically valid and semantically faithful within the MATHLIB ecosystem.

Our approach proceeds in two stages. First, we establish a MATHLIB-native foundational strategy that grounds geometry in $\mathbb{R}^2$ inner product spaces, ensuring compatibility with the broader mathematical library (Section 3.1). Second, we introduce a structure-aware generation pipeline that progressively transforms implicit intuitions into explicit formal constraints through reasoning and retrieval (Section 3.2).

## 3.1. Geometric Formalization Strategy

While recent works such as LeanEuclid (System E) have successfully automated Euclidean geometry through a synthetic, axiomatic framework, we argue that a robust automated mathematician must operate within the standard formalism of modern mathematics rather than isolated logical subsystems. The divergence between our framework and System E stems fundamentally from the choice between synthetic and analytic foundations, which dictates the system's extensibility, rigor, and ecosystem integration.

**Foundational Ontology and Ecosystem Integration.**
System E models geometry using opaque primitive types (e.g., `Point`, `Line`, `Circle`) and atomic predicates (e.g., `Between`, `Collinear`, `SameSide`) governed by custom axioms. While this simplifies formalization by mimicking human intuition, it creates a "logical island" that is structurally incompatible with Lean's broader mathematical library (MATHLIB). In contrast, our method adopts an analytic geometry approach native to MATHLIB, formalizing the plane as a Euclidean space over real numbers ($\mathbb{R}^2$), specifically defined as

```
abbrev Plane := EuclideanSpace ℝ (Fin 2)
```

This grounds geometry in vector spaces, enabling direct invocation of MATHLIB's vast ecosystem—from `MeasureTheory` to `Analysis`—allowing operations like derivatives and complex numbers that are fundamentally inaccessible to axiomatic DSLs.

**Explicit Rigor versus Implicit Intuition.** A critical distinction lies in the handling of "diagrammatic gaps." System E delegates "obvious" geometric truths, such as point distinctness and line intersections, to external SMT solvers. While this simplifies the generative task, it reduces reasoning transparency and hides logical steps. In contrast, native MATHLIB enforces kernel-verified explicit rigor. Our approach compels the language model to identify these necessary implicit non-degeneracy conditions explicitly. Although this imposes a stricter generative requirement by treating implicit assumptions as rigid syntax barriers, it ensures that proofs are mathematically complete, self-contained, and verifiable solely within the standard Lean kernel without reliance on external oracles.

**Case Study: Circle Intersections.** We illustrate the distinction using the formalization of circle intersections—the existence of a common point between two intersecting circles. System E introduces custom predicates (e.g., `Circle`, `intersectsCircle`, `onCircle`) and relies on a custom axiom where the existence precondition (`intersectsCircle`) is verified externally by SMT, keeping the proof script sparse but opaque:

```
-- Builds on LeanEuclid's System E
-- Custom predicates: Circle, intersectsCircle,
onCircle
import LeanEuclid.SystemE

-- System E: Custom axiom with external SMT check
axiom intersection_circles (α β : Circle)
  -- Validity check delegated to SMT
  (h : intersectsCircle α β) :
  exists c : Point, (onCircle c α) ∧ (onCircle c β)

-- Usage requires diagrammatic reasoning
theorem prop_1 : forall (a b : Point) (AB : Line),
  distinctPointsOnLine a b AB →
  exists c : Point,
    |(c--a)| = |(a--b)| ∧ |(c--b)| = |(a--b)| := by
  euclid_intros
  euclid_apply circle_from_points a b as BCD
  euclid_apply circle_from_points b a as ACE
  -- SMT proves: intersectsCircle BCD ACE
  euclid_apply intersection_circles BCD ACE as c
  euclid_finish
```

Conversely, native MATHLIB grounds geometry in analytic definitions over $\mathbb{R}^2$, enforcing explicit algebraic bounds, such as positivity and triangle inequalities, as compilation guards to ensure full kernel-level verification:

```
-- Standard Mathlib only, no custom axioms
import Mathlib
abbrev Plane := EuclideanSpace ℝ (Fin 2)

-- Native Mathlib: Explicit non-degeneracy guards
required
theorem circle_intersection
  (O1 O2 : EuclideanSpace ℝ (Fin 2)) (r1 r2 : ℝ)
  -- Explicit Positivity
  (hr : r1 > 0 ∧ r2 > 0)
  -- Explicit Triangle Inequality
  (h_tri : dist O1 O2 < r1 + r2)
  -- Explicit Inclusion Check
  (h_inc : dist O1 O2 > abs (r1 - r2))
  : ∃ P, P ∈ Metric.sphere O1 r1 ∧ P ∈ Metric.sphere
O2 r2 := by
  -- Proof requires algebraic manipulation
  sorry
```

## 3.2. Formalization Generation Pipeline

The fundamental challenge in automated geometry formalization is the *semantic misalignment* between natural language and formal logic. Natural language geometry relies heavily on *implicit assumptions* derived from spatial intuition, whereas MATHLIB's analytic geometry demands *explicit specifications* grounded in rigorous type theory. To bridge this gap and ensure native interoperability with the MATHLIB ecosystem, we propose a modular framework composed of four stages: (1) *Constraint Explication*, (2) *Reasoning-based Configuration Anchoring*, (3) *Formalization Mapping*, and (4) *Iterative Repair*. Figure 1 illustrates this architecture.

### 3.2.1. CONSTRAINT EXPLICATION: FROM IMPLICIT INTUITION TO EXPLICIT RIGOR

Natural language problem statements are inherently underspecified, omitting preconditions that are technically re-

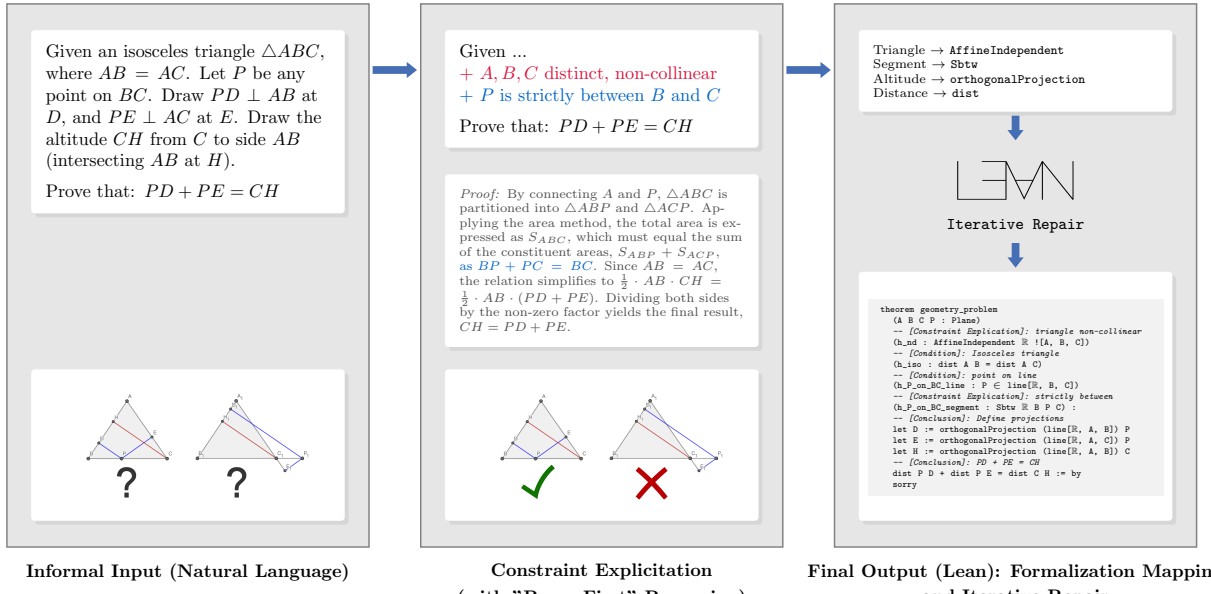

*Figure 1.* Overview of the EUCLEAN formalization pipeline. **Left** (Ambiguity in Input): The process begins with an informal natural language statement containing topological ambiguity—for instance, "Point $P$ on $BC$" could topologically imply a segment, ray, or line. The diagrams illustrate that while the left configuration is the intended solution, the right configuration (on the extension) is invalid for the problem's logic. **Middle** (Constraint Explication): The system employs a "Prove-First" strategy to resolve these ambiguities. The generated intermediate proof relies on area summation ($S_{ABC} = S_{ABP} + S_{ACP}$), a logical dependency that holds only if $P$ is strictly within the segment $BC$. This reasoning "anchors" the configuration, forcing the system to explicitly inject non-degeneracy and topological conditions. **Right** (Formalization Mapping & Repair): These explicated constraints are retrieved and mapped to canonical MATHLIB primitives (e.g., mapping "Triangle" to `AffineIndependent` constraints). Finally, the system utilizes an iterative repair loop based on Lean compiler feedback to synthesize a rigorous, type-checked theorem statement. Overall, this supports reliable MATHLIB-native formalization.

quired for mathematical well-definedness. In the analytic framework of MATHLIB, these implicit assumptions must be transformed into explicit hypotheses. Our framework systematically injects two categories of constraints:

**Explicating Topological Configurations:** Vague positional descriptions such as "point $D$ lies on segment $BC$" imply strictly more than collinearity; they carry implicit relative positional constraints. Our system translates these topological intuitions into explicit conditions (e.g., converting "on segment" to `Sbtw ℝ B D C` for strict betweenness), thereby grounding diagrammatic intuition in precise vector space formalism.

**Explicating Non-degeneracy:** Geometric functions in MATHLIB are often partial or undefined in degenerate cases. While a text might simply reference "triangle $ABC$," formal validity in MATHLIB requires an explicit hypothesis of affine independence (e.g., `AffineIndependent ℝ ![A, B, C]`). Similarly, defining "line $AB$" implicitly presupposes distinct points ($A \neq B$), and a "circle" presupposes a positive radius ($r > 0$). Our pipeline instantiates these latent conditions to prevent

the generation of ill-typed or vacuous statements.

### 3.2.2. CONFIGURATION ANCHORING VIA "PROVE-FIRST" REASONING

Direct translation often fails to recover these implicit constraints due to a lack of context. To address this, we employ a "Prove-First, Formalize-Second" strategy. We prompt the model to first generate an informal proof sketch. This process forces the model to resolve ambiguities and commit to a specific geometric configuration (e.g., determining point ordering or tangency orientation). This intermediate reasoning serves as a configurational anchor, significantly improving the model's ability to identify and explicate hypotheses that were unstated in the original problem text.

### 3.2.3. FORMALIZATION MAPPING VIA RETRIEVAL

To ensure compatibility with MATHLIB's standard library, we implement a *Geometry*-MATHLIB *Alignment* mechanism. This module acts as a semantic bridge, mapping informal geometric entities to their canonical MATHLIB representations through type promotion.

As shown in Table 1, our system does not merely transliterate terms; it retrieves rigorous definitions and automatically associates the necessary explicit constraints. For instance, mapping an informal "angle" to `EuclideanGeometry.angle` triggers the injection of nonzero vector preconditions required by inner product spaces.

### 3.2.4. FEEDBACK-DRIVEN ITERATIVE REPAIR

Finally, to satisfy the strict syntactic requirements of Lean 4, we employ a feedback-driven repair loop. The generated formal statement is verified by the Lean compiler; if compilation fails, the specific error messages are fed back to the model. This closes the generation loop, allowing the model to iteratively refine its explicit specifications against the ground-truth logic of the proof assistant.

## 4. Datasets

We apply EUCLEAN to construct two geometry formalization datasets of different scales and characteristics. We use the DeepSeek-V3 model (DeepSeek-AI, 2024) as the generator throughout the pipeline stages described in Section 3.2.

### 4.1. OMNI-Geometry

OMNI-Geometry is derived from the OMNI-Math dataset (Gao et al., 2025), focusing specifically on plane geometry problems. We filter problems by domain to obtain 780 geometry problems spanning a comprehensive difficulty range from 1.0 to 9.5 on a 10-point scale, with sources including IMO, USAMO, national olympiads, and regional competitions.

For each problem, we generate 32 candidate formalizations. Each candidate is produced by the complete four-stage pipeline in Section 3.2. We retain 768 problems (98.5%) with at least one compiling formal statement. The high retention rate reflects the effectiveness of the iterative repair stage.

**Example Formalization.** Figure 2 presents a complete formalization from OMNI-Geometry, illustrating the structure of our outputs. The example highlights two key capabilities of EUCLEAN for bridging informal problem statements and the strict requirements of MATHLIB. First, for *topological constraint explication*, the phrase "$D$, $A$, and $B$ are collinear in that order" encodes ordering information that is crucial for the intended (unique) configuration; our system recognizes this and maps it to the precise MATHLIB predicate `Sbtw ℝ D A B` (strict betweenness), thereby excluding other collinear permutations. Second, for *non-degeneracy guarding*, although the problem simply mentions equilateral triangles, strict formalization requires safeguarding against trivial edge cases (e.g., points collapsing

```
/-- HMMT_11_794:
$H O W, B O W$, and $D A H$ are equilateral
triangles in a plane such that $W O=7$ and
$A H=2$. Given that $D, A, B$ are collinear
in that order, find the length of $B A$.
Prove that the answer/conclusion is: 11. --/
theorem equilateral_triangles_problem
    (H O W B D A : Plane)
    -- [Constraint Explication] Triangle non-
degeneracy
    (h_HOW_affine_independent : AffineIndependent ℝ
![H, O, W])
    (h_BOW_affine_independent : AffineIndependent ℝ
![B, O, W])
    (h_DAH_affine_independent : AffineIndependent ℝ
![D, A, H])
    -- [Condition] Equilateral triangle
    (h_HOW_equilateral : dist H O = dist O W ∧ dist O
W = dist W H ∧ dist W H = dist H O)
    (h_BOW_equilateral : dist B O = dist O W ∧ dist O
W = dist W B ∧ dist W B = dist B O)
    (h_DAH_equilateral : dist D A = dist A H ∧ dist A
H = dist H D ∧ dist H D = dist D A)
    -- [Condition] Given lengths
    (h_WO_eq_7 : dist W O = 7)
    (h_AH_eq_2 : dist A H = 2)
    -- [Condition] Collinearity condition
    (h_DAB_collinear : Collinear ℝ {D, A, B})
    -- [Constraint Explication] Relative positioning
    (h_DAB_order : Sbtw ℝ D A B) :
    dist B A = 11 := by
```

*Figure 2.* Example formalization from OMNI-Geometry (HMMT 2011 Problem 794). It illustrates how EUCLEAN makes implicit geometric information explicit by (i) translating topological ordering into a strict betweenness constraint and (ii) adding non-degeneracy guards required to make triangles well-defined in MATHLIB. This formalization is proved in Appendix G.

onto a line). Accordingly, the pipeline automatically injects `AffineIndependent` hypotheses to ensure the geometric entities represent the intended non-degenerate figures.

### 4.2. Numina-Geometry

Numina-Geometry is constructed from the NuminaMath dataset (LI et al., 2024), which aggregates mathematical problems from diverse sources including textbooks, competitions, and online resources.[2] We filter for geometry-related problems using the `problem_type == Geometry` criterion, obtaining 183,796 candidate problems.

We apply the same complete pipeline as in Section 3.2 to this larger corpus. After formalization and compiler-guided iterative repair, we retain 177,597 problems (96.6%) with at least one compiling formal statement. This large-scale dataset provides extensive coverage of geometry problems at varying difficulty levels, from elementary constructions to advanced competition problems.

---

[2]The official NuminaMath-LEAN dataset explicitly "filter[s] out geometry and combinatorics problems, because of their difficulty of formalization" (AI-MO, 2025). Our work addresses this gap by providing geometry formalizations for the excluded subset.

*Table 1.* Examples of mappings from implicit informal concepts to explicit MATHLIB-native representations. The third column highlights the implicit constraints (e.g., non-degeneracy, relative positioning) that EUCLEAN automatically injects to ensure mathematical validity.

| Informal Concept | MATHLIB Representation | Auto-Explicated Constraints |
|---|---|---|
| Line $AB$ | `line[ℝ, A, B]` | `A ≠ B`
*(Distinctness condition)* |
| Angle $\angle ABC$ | `angle A B C` | `A ≠ B ∧ C ≠ B`
*(Well-definedness condition)* |
| Triangle $ABC$ | `![A, B, C]` | `AffineIndependent ℝ ![A, B, C]`
*(Implies non-collinear & distinct)* |
| $P$ on Segment $AB$
$(A - P - B)$ | `P ∈ segment ℝ A B`
*(implied by constraint)* | `Sbtw ℝ A P B`
*(Strict betweenness; implies membership)* |
| Circle $\odot(O, r)$ | `sphere O r` | `r > 0`
*(Positive radius)* |
| Tangency
(Line-Circle) | `dist O (orthogonalProjection`
`(line[ℝ, A, B]) O) = r` | `A ≠ B ∧ r > 0` |
| $A, B, C$ on $\odot(O, r)$
$(A - B - C$ CCW$)$ | `dist A O = r ∧ dist B O = r`
`∧ dist C O = r` | `(B-A) 0 * (C-A) 1 - (B-A) 1 * (C-A) 0 > 0`
*(Cross Product)* |

*Table 2.* Dataset statistics. Both datasets are generated by the complete four-stage pipeline in Section 3.2.

| | OMNI-Geo | Numina-Geo |
|---|---|---|
| Initial problems | 780 | 183,796 |
| Valid formalizations | 768 | 177,597 |
| Retention rate | 98.5% | 96.6% |

# 5. Experiments

We evaluate EUCLEAN through three sets of experiments: ablation studies on our formalization pipeline, human evaluation of formalization quality, and downstream training experiments on neural theorem proving.

## 5.1. Ablation Study

We conduct ablation experiments on 30 OMNI-Geometry problems to evaluate the contribution of each component in our formalization pipeline. For each configuration, we sample 32 independent generations per problem, resulting in 960 candidate formalizations per configuration.

**Setup.** We compare five configurations: (1) Basic prompting with minimal instructions, (2) adding formalization mapping of concepts, (3) adding iterative code repair, (4) adding formalization mapping of constraints, and (5) adding "Prove-First" Reasoning (configuration anchoring).

**Results.** Table 3 shows the number of compiling candidates out of 960 generations for each configuration. The basic prompt achieves 125 compiling candidates (13.0%).

Adding concept mapping more than doubles this to 254 (26.5%), demonstrating the importance of retrieving MATHLIB-native primitives and type signatures. Iterative code repair provides the largest individual gain, reaching 465 (48.4%), as it allows recovery from common type and syntax errors. Adding explicit constraint mapping further improves compilation to 500 (52.1%).

Interestingly, adding "Prove-First" configuration anchoring slightly reduces the count to 480 (50.0%). This is expected: compilation is a scalable proxy rather than a semantic metric. Anchoring commits the model to a concrete geometric configuration and thus encourages additional configuration constraints (e.g., strict betweenness or sidedness) that are often omitted in minimal statements. Missing these extra constraints would not necessarily cause a Lean compilation failure, so the compilation metric can decrease even though the resulting statements are closer to the intended configuration. To directly assess this effect, we additionally ran a human Pass@3 paired evaluation on the 30 ablation problems. Anchoring wins in 7 cases and loses in 3 cases (10 both correct, 10 both incorrect), suggesting a net improvement in semantic faithfulness despite the lower raw compilation count.

## 5.2. Human Evaluation

To assess the semantic quality of our formalizations beyond automated filtering, we conduct human evaluation with expert annotators.

**Setup.** We randomly sample 180 problems from the subset of OMNI-Geometry with a difficulty score of 4.0 or higher. For each problem, we prepare 5 compiling formalizations

*Table 3.* Ablation study on pipeline components. Numbers indicate compiling candidates out of 960 generations per configuration. Most components improve compilation, while configuration anchoring slightly lowers compilation by adding explicit geometric constraints; its semantic effect is evaluated separately by human annotation.

| Configuration | Compiling |
|---|---|
| Basic prompting | 125 (13.0%) |
| + concepts formalization mapping | 254 (26.5%) |
| + iterative code repair | 465 (48.4%) |
| + constraints formalization mapping | 500 (52.1%) |
| + configuration anchoring | 480 (50.0%) |

*Table 4.* Human evaluation results by difficulty bucket on 180 OMNI-Geometry problems. TOP1/TOP5 are reported as #correct (accuracy). Multi-sample generation substantially improves the chance of semantic faithfulness.

| Difficulty | Total | TOP1 | TOP5 |
|---|---|---|---|
| $4.0 - 5.0$ | 53 | 27 (50.94%) | 41 (77.36%) |
| $5.0 - 6.0$ | 77 | 43 (55.84%) | 59 (76.62%) |
| $6.0 - 7.0$ | 16 | 7 (43.75%) | 12 (75.00%) |
| $7.0 - 8.0$ | 19 | 8 (42.11%) | 11 (57.89%) |
| $8.0+$ | 15 | 3 (20.00%) | 9 (60.00%) |
| Overall $(4.0+)$ | 180 | 88 (48.89%) | 132 (73.33%) |

produced by our pipeline. Annotators label whether each formalization is semantically consistent with the original problem statement. We report TOP1 accuracy (whether the first formalization is correct) and TOP5 accuracy (whether at least one of the five formalizations is correct). Annotators are graduate students with experience in formal mathematics and LEAN 4.

**Results.** Table 4 presents results broken down by detailed difficulty intervals. Performance is strongest in the $4.0 - 6.0$ range, where TOP1 accuracy consistently exceeds 50% and TOP5 accuracy maintains approximately 77%. We observe a gradual decline as difficulty increases: in the $7.0 - 8.0$ range, TOP1 accuracy dips to 42.11% and TOP5 to 57.89%. Notably, for the most challenging problems (difficulty 8.0+), while TOP1 accuracy decreases to 20.00%, the TOP5 accuracy remains resilient at 60.00%.

Overall, the TOP1 accuracy is 48.89% and the TOP5 accuracy is 73.33%. The consistent gap between TOP1 and TOP5 across all difficulty levels indicates that sampling multiple candidates substantially increases the chance of obtaining a semantically faithful formalization. These results indicate that the pipeline is reliable for many problems under multi-sample generation, but hard instances remain challenging and account for a disproportionate share of semantic mismatches.

## 5.3. Inference and Iteration Training

To assess the ecosystem compatibility and downstream utility of our formalized datasets, we conduct inference and training experiments on the Goedel v2 model (Lin et al., 2025).

**Setup.** We use Goedel v2, an 8-billion parameter language model fine-tuned for theorem proving in LEAN 4. Starting from the base model, we generate proof attempts for all problems in our Numina-Geometry dataset. Compiling proofs are collected to construct a training corpus. We investigate two distinct training strategies starting from the base checkpoint to assess data efficiency: (1) Supervised Fine-Tuning (SFT), utilizing all successful proof traces (collected from 1 generation attempt per problem); and (2) Direct Preference Optimization (DPO), utilizing pairs of (successful, failed) attempts for the same theorem (collected from 2 generation attempts per problem).

**Evaluation.** We evaluate on the full Numina-Geometry dataset (177,597 formalized problems). We report the Pass@1 success rate, measuring the percentage of problems solved with a single generation attempt.

**Results.** Table 5 presents the results. Crucially, the base Goedel v2 model achieves a 13.6% proof success rate on Numina-Geometry without any geometry-specific training. This result highlights the significant advantage of our MATH-LIB-native approach: unlike domain-specific languages that require specialized solvers or extensive retraining, our formalization enables knowledge transfer and is immediately accessible to general-purpose theorem provers. Further training on our dataset improves the success rate to 15.1%, validating the quality of our formalizations and their utility for advancing unified neural theorem proving. Additional held-out and cross-prover results are reported in Appendix D.

**Analysis of Modest Improvement.** The relatively modest gain from training (13.6% $\rightarrow$ 15.1%) can be attributed to the scarcity of geometry problems in existing theorem proving corpora. Since general-purpose provers like Goedel v2 are predominantly trained on algebraic and number-theoretic problems, they generate few successful proofs on geometry, resulting in limited positive samples for SFT and limited preference pairs for DPO. Additional held-out OMNI-Geometry and DeepSeek-Prover-V2-7B results in Appendix D show the same modest positive trend, suggesting that the downstream signal is not merely a single-prover or in-distribution artifact. To assess the potential of our dataset with frontier models, we evaluated Aristotle (The Harmonic Team, 2025)—a state-of-the-art neural theorem prover—on 100 randomly sampled problems. After filtering

*Table 5.* Inference and iteration training results on Numina-Geometry. This demonstrates transferability to downstream provers.

| Model | Pass@1 (%) |
|---|---|
| Goedel v2 (base) | 13.6 |
| base + SFT (round 1 passed) | 15.1 |
| base + DPO (pass/fail pairs) | 15.0 |

cases that Aristotle explicitly judged false and refuted, cases that became trivial due to obvious misformalization, and cases with concrete counterexamples, Aristotle still proves 25 problems, while another 19 remain unresolved but unrefuted. This suggests a rough but informative lower-bound signal of the corpus's semantic quality. Detailed successful proofs are presented in Appendix G. In future work, such frontier models can be leveraged to generate high-quality proof traces at scale, enabling more effective training on our geometry formalization dataset.

## 6. Discussion and Limitations

**Advantages of Unified Verification.** By formalizing geometry problems in LEAN 4 with MATHLIB, we achieve several benefits over domain-specific approaches. First, proofs are verified by an established, extensively tested proof assistant, minimizing trusted computing base. Second, formalized statements can leverage the full power of MATHLIB's mathematical infrastructure, including topology, analysis, and algebra when needed for proofs. Third, our formalizations integrate with the broader neural theorem proving ecosystem, enabling training and evaluation using standard tools and benchmarks.

**Limitations.** Our approach has several limitations. First, semantic validation remains difficult at corpus scale. Geometry formalization often requires judging whether added or omitted configuration assumptions preserve the intended problem, yet scalable, high-quality labels for this semantic check are unavailable. We therefore view Numina-Geometry as a large-scale MATHLIB-native candidate corpus with layered validation, to be further refined by prover-based filtering and expert review. Second, the expressiveness of our formalization is bounded by MATHLIB's current geometry coverage, which may not include all constructions used in competition problems. Third, our pipeline relies on the LLM to identify and explicate implicit geometric constraints; subtle necessary conditions may be overlooked, leading to under-specified formal statements. Conversely, explicitly injecting non-degeneracy conditions to guard against all potential edge cases may introduce redundant or over-strong hypotheses, thereby reducing theorem generality even when the properties hold more broadly.

**Text-Only Formalization.** Unlike multimodal approaches such as MATP-BENCH (He et al., 2025), our framework operates solely on textual problem descriptions and does not process geometric diagrams. This design choice requires that input problems contain complete and precise natural language specifications without implicit information encoded in figures. While this may seem limiting, we argue it aligns with the foundational principles of formal mathematics. As Hilbert emphasized in his axiomatization of geometry, all relevant properties must be explicitly stated rather than inferred from visual intuition. Modern mathematical olympiads increasingly adopt this practice—the IMO Grand Challenge explicitly requires that problem statements be "fully specified in text" to enable rigorous formalization.[3] Our text-only approach thus focuses on the mathematically rigorous subset of geometry problems amenable to formal verification, leaving multimodal diagram parsing to future work.

**Broader Impact.** Our datasets and methodology contribute to the development of reliable AI systems for mathematics. By grounding geometry reasoning in verified formal systems, we reduce the risk of subtle errors that could propagate in downstream applications. However, we note that our formalized statements should not be considered ground truth without expert review, as our pipeline has imperfect accuracy.

## 7. Conclusion

We presented EUCLEAN, a systematic framework that bridges the gap between informal geometry problems and the rigorous requirements of LEAN 4's MATHLIB. By combining a constraint-aware formalization pipeline with iterative repair, we constructed two large-scale datasets: OMNI-Geometry (768 competition-level problems) and Numina-Geometry (177,597 problems). Our experiments, spanning human evaluation and downstream prover training, validate the quality of these resources and demonstrate that they enable general-purpose neural theorem provers to extend their capabilities to geometry tasks within standard MATHLIB, offering a unified alternative to domain-specific solvers.

Our work takes a step toward unifying geometry reasoning within established formal verification ecosystems. Future directions include extending the system's coverage to analytic and solid geometry, refining the balance between constraint sufficiency and generality, and leveraging frontier theorem provers to generate high-quality proof traces for more effective model training.

---

[3] See discussion at https://leanprover.zulipchat.com/#narrow/channel/208328-IMO-grand-challenge/topic/A.20story.20with.20nondegeneracy.20conditions.

## Impact Statement

This paper presents work whose goal is to advance the field of machine learning, specifically in automated theorem proving and mathematical formalization. Our framework enables the translation of informal geometry problems into formally verified specifications, contributing to the development of more reliable AI systems for mathematics. By grounding reasoning in established proof assistants, we reduce the risk of subtle mathematical errors that could propagate in downstream applications. The datasets and methodology we release are intended to support research in neural theorem proving. We do not foresee immediate negative societal consequences beyond those common to advances in automated reasoning systems.

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

# A. Lean Version and Fixed Header

We report the Lean toolchain version and the fixed header used throughout our MATHLIB-native geometry formalizations.

```
-- Lean version:
-- leanprover/lean4:v4.17.0-rc1

import Mathlib
open Real EuclideanGeometry Metric MeasureTheory
Affine AffineSubspace Triangle Finset
open scoped Real RealInnerProductSpace BigOperators
abbrev Plane := EuclideanSpace ℝ (Fin 2)
```

# B. Dataset Statistics

Table 6 presents the distribution of our OMNI-Geometry problems by difficulty level. The dataset includes problems from various competition sources, with HMMT (Harvard-MIT Mathematics Tournament) contributing 557 problems—the largest single source in our competition-level dataset.

*Table 6.* OMNI-Geometry difficulty distribution.

| Difficulty Range | Problem Count |
| --- | --- |
| 1.0–2.0 | 19 (2.5%) |
| 2.0–3.0 | 48 (6.3%) |
| 3.0–4.0 | 44 (5.7%) |
| 4.0–5.0 | 222 (28.9%) |
| 5.0–6.0 | 251 (32.7%) |
| 6.0–7.0 | 60 (7.8%) |
| 7.0–8.0 | 67 (8.7%) |
| 8.0+ | 57 (7.4%) |
| TOTAL | 768 (100.0%) |

For Numina-Geometry, Numina metadata provide source-level information. Table 7 reports this distribution, and Tables 8 and 9 summarize representative formal-side coverage and a rough difficulty proxy based on hypothesis complexity.

*Table 7.* Source distribution of Numina-Geometry.

| Source | Fraction |
| --- | --- |
| olympiads | 32.51% |
| cn_k12 | 26.15% |
| synthetic_math | 17.75% |
| aops_forum | 11.01% |
| Others | 12.58% |

# C. Training Hyperparameters

We conduct training experiments on NVIDIA A100-40GB GPUs. Table 10 summarizes the hyperparameters for SFT and DPO training.

*Table 8.* Representative primitive coverage in Numina-Geometry. This is not an exhaustive vocabulary count, but a formal-side proxy for dataset coverage.

| Bucket | Representative constructs |
|---|---|
| Non-degeneracy | `AffineIndependent` (47.52%) |
| Incidence / order | `line` (19.86%), `segment` (12.71%), `midpoint` (14.42%), `Collinear` (9.05%), `Sbtw/Wbtw` (4.81%) |
| Angles / relations | `angle` (35.44%), `orthogonalProjection` (15.29%), `tangent` (9.24%), `parallel` (11.76%) |
| Circle | `sphere/circumsphere` (20.62%) |
| Centers | `circumcenter` (5.00%), `incenter/inradius` (3.85%) |
| Area | `convexHull` (20.19%), `volume` (17.54%) |

*Table 9.* Distribution of proposition-like `h_*` hypotheses in Numina-Geometry. This provides a rough proxy for the difficulty distribution, since problems requiring more explicit hypotheses tend to involve more constraints and configuration details.

| # of `h_*` hypotheses | Fraction |
|---|---|
| 0–4 | 38.60% |
| 5–9 | 36.77% |
| 10–14 | 15.86% |
| 15–19 | 5.98% |
| 20+ | 2.79% |

*Table 10.* Training hyperparameters for SFT and DPO experiments.

| Hyperparameter | SFT | DPO |
|---|---|---|
| Base model | Goedel v2 (8B) | Goedel v2 (8B) |
| Number of nodes | 1 | 1 |
| GPUs per node | 8 | 8 |
| Total A100 GPUs | 8 | 8 |
| Learning rate | $1 \times 10^{-4}$ | $1 \times 10^{-5}$ |
| DPO $\beta$ | – | 0.1 |
| Batch size (per device) | 1 | 1 |
| Gradient accumulation | 32 | 32 |
| Effective batch size | 256 | 256 |
| Number of epochs | 1 | 1 |
| Max sequence length | 4096 | 2048 |
| LR scheduler | Cosine | Cosine |
| Warmup ratio | 0.05 | 0.05 |
| Precision | BF16 | BF16 |
| DeepSpeed | ZeRO-3 | ZeRO-3 |
| Flash Attention | v2 | v2 |

*Table 11.* Held-out OMNI-Geometry proof success.

| Metric | Base | SFT | DPO |
|---|---|---|---|
| Pass@1 | 6.90% | 7.68% | 7.81% |
| Pass@2 | 8.33% | 9.11% | 8.59% |

## D. Additional Downstream Utility Results

**Held-out OMNI-Geometry.** OMNI-Geometry is not used in any training stage. Table 11 shows that training on Numina-Geometry transfers to this held-out and harder benchmark.

**Goedel v2 stability.** Two independent base inference runs achieve 13.62% and 13.58% Pass@1. SFT and DPO achieve 15.11% and 15.02%, respectively. Paired gain/loss counts are 5,635/2,988 for SFT and 5,338/2,853 for DPO; DPO uses 7,148 preference pairs. These counts suggest that the gain is not merely sampling variance, although the absolute improvement remains modest.

**DeepSeek-Prover-V2.** To check whether the downstream signal is specific to Goedel v2, we also run SFT with DeepSeek-Prover-V2-7B (Ren et al., 2025). We collect successful proof traces on a randomly sampled half of Numina-Geometry using the base model itself, then train for 2 epochs with the same SFT recipe except for the epoch count. Evaluation is Pass@1 on the same half-corpus split.

The 1.6-point improvement is consistent with Goedel v2's 13.6% to 15.1% gain, indicating that the Numina-Geometry training signal is not an artifact of a single prover. The modest absolute gain also supports our diagnosis that the current bottleneck is the quantity and quality of geometry proof traces available from open-source provers.

## E. AlphaGeometry Soundness Issues

AlphaGeometry's domain-specific language has documented soundness issues discussed in the Lean theorem proving community (Lean Prover Community, 2025):

1. **Orientation Checking.** Orientation is determined by numerically generated diagrams without rigorous mathematical foundation. The correctness of proofs may depend on random choices during diagram generation.

2. **Incorrect Inference Rules.** The deductive database contains invalid rules. For example, the rule `cyclic A B P Q` $\Rightarrow$ `eqangle P A P B Q A Q B` is mathematically incorrect in general configurations.

3. **Non-degeneracy Conditions.** Conditions like non-collinearity are checked by inspecting an internal graph representation rather than through formal reasoning, which is significantly more challenging and error-prone.

These issues underscore the need for geometry reasoning grounded in verified proof assistants like Lean with MATH-LIB.

## F. Comparison with Related Benchmarks

We provide detailed comparisons with concurrent geometry formalization efforts, highlighting how EUCLEAN achieves

*Table 12.* Additional SFT result with DeepSeek-Prover-V2-7B.

| Model | Pass@1 |
|---|---|
| DeepSeek-Prover-V2-7B (base) | 13.4% |
| + SFT | 15.0% |

full MATHLIB compatibility without custom axiom systems or external SMT solvers.

*Table 13.* Comparison of geometry formalization approaches. This highlights EUCLEAN's scalability and ecosystem compatibility.

| | EUCLEAN | LeanEuclid | LeanGeo | MATP |
|---|---|---|---|---|
| Scale | 178,365 | 173 | ∼200 | 1,056 |
| Prover | Lean 4 | Lean 4 | Lean 4 | Mixed |
| Library | MATHLIB | System E | System E + custom | Mixed |
| SMT Required | No | Yes | Yes | No |
| Custom Axioms | No | Yes | Yes | Partial |
| Auto-generated | Yes | No | No | No |

### F.1. Detailed Code Comparison

We illustrate the fundamental differences between formalization approaches through concrete examples. The key distinction is that EUCLEAN uses only standard MATHLIB definitions and tactics, while LeanEuclid and LeanGeo require custom axiom systems (System E) and SMT solvers for proof automation.

**Problem: Isosceles Triangle Angle Equality.** *In triangle ABC with AB = AC, prove that ∠ABC = ∠ACB.*

**LeanEuclid Formalization** (Custom System E with SMT):

```
-- Requires custom primitives: Point, OnLine, onCircle
-- Uses non-standard axiom system (System E)
import LeanEuclid.SystemE

theorem isoTriangle_eq_angles :
  forall (A B C : Point) (AB AC BC : Line),
  distinctPointsOnLine A B AB →
  distinctPointsOnLine A C AC →
  distinctPointsOnLine B C BC →
  |(A--B)| = |(A--C)| →
  <<B:A:C>> = <<C:A:B>> := by
  euclid_intros
  euclid_apply exists_midpoint B C as D
  euclid_apply line_from_points B C as BC
  -- SMT solver handles diagrammatic reasoning
  euclid_finish  -- Calls CVC5/Z3 SMT solver
```

**LeanGeo Formalization** (Extended System E with Custom Library):

```
-- Builds on LeanEuclid's System E
-- Custom predicates: IsoTriangle, angle notation
import LeanGeo.Library

theorem isoTriangle_imp_eq_angles :
  forall (A B C : Point),
  IsoTriangle A B C →
  ANG A:B:C = ANG A:C:B := by
  euclid_intros
  euclid_apply exists_midpoint B C as D
  euclid_apply line_from_points B C as BC
  euclid_apply coll_angles_eq
```

```
  euclid_apply congruentTriangles_SSS D B A D C A
  euclid_apply coll_angles_eq
  euclid_finish  -- SMT-based reasoning via LeanSMT
```

**EUCLEAN Formalization** (Pure MATHLIB):

```
-- Standard Mathlib only, no custom axioms
import Mathlib
abbrev Plane := EuclideanSpace ℝ (Fin 2)

theorem isosceles_triangle_angles
  (A B C : Plane)
  (hnd : ¬ (Collinear ℝ {A, B, C}))
  (hiso : dist A B = dist A C) :
  EuclideanGeometry.angle A B C =
  EuclideanGeometry.angle A C B := by
  -- Uses standard Mathlib tactics
  exact EuclideanGeometry.angle_eq_angle_of_dist_eq
hiso
```

### F.2. Key Differences Summary

*Table 14.* Comparison of formalization primitives.

| Concept | LeanEuclid/LeanGeo | EUCLEAN (MATHLIB) |
|---|---|---|
| Point | Point (custom) | EuclideanSpace ℝ (Fin 2) |
| Line | Line (custom) | AffineSubspace.line |
| Circle | Circle (custom) | Metric.sphere O r |
| On Line | OnLine p L | p ∈ AffineSubspace.line |
| On Circle | onCircle p C | p ∈ Metric.sphere O r |
| Angle | <<A:B:C>> | EuclideanGeometry.angle |
| Distance | |(a--b)| | dist A B |
| Midpoint | exists_midpoint | midpoint ℝ A B |
| Proof | SMT | Standard Lean tactics |
| | euclid_finish | |

**Advantages of EUCLEAN's MATHLIB-Native Approach.**

- **Sound foundations**: All proofs are verified by Lean's kernel without external SMT oracles.

- **Ecosystem compatibility**: Works with existing MATHLIB theorems in analysis, algebra, and topology.

- **Neural prover training**: Compatible with models trained on standard MATHLIB (e.g., Goedel Prover, DeepSeek-Prover).

- **No trusted code**: Does not require believing custom axiom systems or SMT solver correctness.

- **Scalability**: Automatic formalization at scale (178K problems vs. hundreds for manual approaches).

**Comparison with MATP-BENCH.** MATP-BENCH (He et al., 2025) is a multimodal benchmark that includes geometric diagrams alongside problem statements, enabling evaluation of visual reasoning capabilities. However, MATP-BENCH contains only 1,056 problems (with geometry being a subset), manually curated and formalized. Our Numina-Geometry dataset provides 177,597 automatically

formalized problems for large-scale training, while OMNI-Geometry (768 problems) focuses on competition-level geometry with multi-stage quality filtering.

## G. Case Study: Proving OMNI-Geometry Examples with Aristotle

We evaluate Aristotle (The Harmonic Team, 2025), a state-of-the-art neural theorem prover, on randomly sampled formalizations from our corpus. In the expanded 100-problem evaluation discussed in Section 5.3, Aristotle proves 25 problems after excluding cases it explicitly refutes as false, trivial misformalizations, and cases with concrete counterexamples; another 19 remain unresolved but unrefuted. Below we present five successful OMNI-Geometry proofs as qualitative examples, showing the problem ID, informal statement, formal Lean theorem statement, and complete Lean proof.

**HMMT_11_206.** In isosceles $\triangle ABC$ with $AB = AC$ and $P$ on segment $BC$, if $\angle BAP = 2\angle CAP$, $BP = \sqrt{3}$, and $CP = 1$, compute $AP$.

```
theorem isosceles_triangle_with_point
    (A B C P : Plane)
    (h_affine : AffineIndependent ℝ ![A, B, C])
    (h_isosceles : dist A B = dist A C)
    (h_P_on_BC : P ∈ segment ℝ B C)
    (h_angle : angle B A P = 2 * angle C A P)
    (h_BP : dist B P = Real.sqrt 3)
    (h_CP : dist C P = 1) :
    dist A P = Real.sqrt 2
```

```
-- HMMT_11_206
noncomputable section AristotleLemmas

/-
Algebraic solution for HMMT 2011 Algebra 206.
-/
lemma hmmt_206_algebraic_sol
    (x y : ℝ)
    (hx : 0 < x)
    (hy : 0 < y)
    (h1 : 1 = x^2 + y^2 - x * y * Real.sqrt 3)
    (h2 : 3 = x^2 + y^2 - x * y)
    (h3 : y^2 - x^2 = Real.sqrt 3) :
    x = Real.sqrt 2 := by
  rw [ eq_comm, Real.sqrt_eq_iff_mul_self_eq_of_pos ]
<;> nlinarith [ mul_pos hx hy, Real.sqrt_nonneg 3,
Real.sq_sqrt ( show 0 ≤ 3 by norm_num ) ]

/-
The ratio of segments BP and CP is equal to the ratio
of the sines of the opposite angles at A.
-/
lemma hmmt_206_sine_rule_ratio
    (A B C P : Plane)
    (h_iso : dist A B = dist A C)
    (h_P_on_BC : P ∈ segment ℝ B C)
    (h_BP_pos : dist B P ≠ 0)
    (h_CP_pos : dist C P ≠ 0)
    (h_affine : AffineIndependent ℝ ![A, B, C]) :
    dist B P / dist C P = Real.sin (angle B A P) /
Real.sin (angle C A P) := by
  -- By the Sine Rule in $\triangle ABP$ and $\
triangle ACP$, we have $\frac{BP}{\sin \angle BAP} = \
frac{AB}{\sin \angle APB}$ and $\frac{CP}{\sin \angle
CAP} = \frac{AC}{\sin \angle APC}$.
  have h_sin_rule_AB : dist B P / Real.sin (
```

```
EuclideanGeometry.angle B A P) = dist A B / Real.sin (
EuclideanGeometry.angle A P B) := by
    rw [ EuclideanGeometry.angle, EuclideanGeometry.
angle ];
    rw [ InnerProductGeometry.angle,
InnerProductGeometry.angle ];
    rw [ Real.sin_arccos, Real.sin_arccos ];
    norm_num [ dist_eq_norm', EuclideanSpace.norm_eq
];
    norm_num [ div_pow, mul_pow, Real.sq_sqrt (
add_nonneg ( sq_nonneg _ ) ( sq_nonneg _ ) ) ];
    rw [ one_sub_div, one_sub_div ];
    · rw [ ← Real.sqrt_div ( by positivity ), ← Real.
sqrt_div ( by positivity ) ];
      congr 1;
      field_simp;
      norm_num [ inner ] ; ring;
    · simp_all +decide [ dist_eq_norm', EuclideanSpace.
norm_eq ];
      constructor <;> intro h <;> simp_all +decide [
Real.sqrt_eq_zero', add_nonneg, sq_nonneg ];
      · norm_num [ show P = A by ext i; fin_cases i
<;> nlinarith! only [ h ] ] at *;
        rw [ affineIndependent_iff_not_collinear ] at
h_affine;
        rw [ collinear_iff_exists_forall_eq_smul_vadd
] at h_affine;
        contrapose! h_affine;
        use B, C - B;
        simp_all +decide [ segment_eq_image ];
        exact ⟨ ⟨ 1, by norm_num ⟩, by obtain ⟨ x, hx,
 rfl ⟩ := h_P_on_BC; exact ⟨ x, by ext i; fin_cases i
<;> norm_num <;> linarith ⟩ );
      · exact h_BP_pos ( by linarith );
    · norm_num [ dist_eq_norm', EuclideanSpace.norm_eq
 ] at *;
      constructor <;> intro h <;> simp_all +decide [
sub_eq_iff_eq_add ];
      · rw [ eq_comm, Real.sqrt_eq_zero' ] at h_iso ;
norm_num [ show C = A by ext i ; fin_cases i <;>
nlinarith! only [ h_iso ] ] at *;
        rw [ affineIndependent_iff_not_collinear ] at
h_affine ; simp_all +decide [ collinear_pair ];
      · norm_num [ show P = A by ext i; fin_cases i
<;> nlinarith! only [ h ] ] at *;
        rw [ affineIndependent_iff_not_collinear ] at
h_affine;
        rw [ collinear_iff_exists_forall_eq_smul_vadd
] at h_affine;
        refine' h_affine ⟨ B, C - B, _ ⟩;
        norm_num [ segment_eq_image ] at *;
        exact ⟨ ⟨ 1, by ext i; fin_cases i <;>
norm_num ⟩, by obtain ⟨ x, hx, rfl ⟩ := h_P_on_BC;
exact ⟨ x, by ext i; fin_cases i <;> norm_num <;>
linarith ⟩ )
  have h_sin_rule_AC : dist C P / Real.sin (
EuclideanGeometry.angle C A P) = dist A C / Real.sin (
EuclideanGeometry.angle A P C) := by
    -- By the Sine Rule in $\triangle ACP$, we have $\
frac{CP}{\sin \angle CAP} = \frac{AC}{\sin \angle APC}
$.
  have h_sin_rule_AC : dist C P / Real.sin (
EuclideanGeometry.angle C A P) = dist A C / Real.sin (
EuclideanGeometry.angle A P C) := by
    have h_sin_rule : ∀ (p1 p2 p3 : Plane), p1 ≠ p2
→ p2 ≠ p3 → p3 ≠ p1 → dist p2 p3 / Real.sin (
EuclideanGeometry.angle p2 p1 p3) = dist p1 p2 / Real.
sin (EuclideanGeometry.angle p1 p3 p2) := by
      intros p1 p2 p3 hp1p2 hp2p3 hp3p1
      have h_sin_rule : Real.sin (EuclideanGeometry.
angle p2 p1 p3) / dist p2 p3 = Real.sin (
EuclideanGeometry.angle p1 p3 p2) / dist p1 p2 := by
        rw [ EuclideanGeometry.angle_comm p2 p1 p3,
EuclideanGeometry.angle_comm p1 p3 p2 ];
        convert EuclideanGeometry.
sin_angle_div_dist_eq_sin_angle_div_dist _ _ using 1;
        · assumption;
        · assumption;
      grind
    by_cases hCA : C = A <;> by_cases hPA : P = A
<;> by_cases hCP : C = P <;> simp_all +decide [
```

```
EuclideanGeometry.angle_comm ];
        · exact dist_comm _ _;
      · grind;
    exact h_sin_rule_AC;
  -- Since $P$ is on segment $BC$, $\angle APB + \
angle APC = \pi$. Thus $\sin(\angle APB) = \sin(\angle
 APC)$.
  have h_sin_eq : Real.sin (EuclideanGeometry.angle A
P B) = Real.sin (EuclideanGeometry.angle A P C) := by
    -- Since $P$ is on segment $BC$, $\angle APB + \
angle APC = \pi$. Thus $\sin(\angle APB) = \sin(\pi -
\angle APC) = \sin(\angle APC)$.
    have h_sum_angle : EuclideanGeometry.angle A P B +
 EuclideanGeometry.angle A P C = Real.pi := by
      -- Since $P$ lies on segment $BC$, we have $\
angle APB + \angle APC = \pi$.
      have h_angle_sum : EuclideanGeometry.angle A P B
 + EuclideanGeometry.angle A P C = Real.pi := by
        have h_linear : P ∈ openSegment ℝ B C := by
          rw [ segment_eq_image ] at h_P_on_BC;
          rcases h_P_on_BC with ⟨ θ, ⟨ hθ₀, hθ₁ ⟩, rfl
 ⟩ ; cases lt_or_eq_of_le hθ₀ <;> cases lt_or_eq_of_le
 hθ₁ <;> simp_all +decide [ openSegment_eq_image ] ;
          · exact ⟨ θ, ⟨ by linarith, by linarith ⟩,
rfl ⟩;
          · aesop
        obtain ⟨ a, b, ha, hb, hab, rfl ⟩ := h_linear;
        rw [ EuclideanGeometry.angle,
EuclideanGeometry.angle ];
        rw [ show B -ᵥ ( a · B + b · C ) = b · ( B - C
) by ext ; simpa using by rw [ ← eq_sub_iff_add_eq' ]
 at hab ; rw [ hab ] ; ring, show C -ᵥ ( a · B + b · C
 ) = a · ( C - B ) by ext ; simpa using by rw [ ←
eq_sub_iff_add_eq' ] at hab ; rw [ hab ] ; ring];
        rw [ show a · ( C - B ) = - ( a · ( B - C ) )
by rw [ ← smul_neg, neg_sub ] ] ; rw [
InnerProductGeometry.angle_neg_right ] ; ring;
        rw [ InnerProductGeometry.
angle_smul_right_of_pos, InnerProductGeometry.
angle_smul_right_of_pos ] <;> linarith;
      exact h_angle_sum;
    rw [ ← Real.sin_pi_sub, ← h_sum_angle,
add_sub_cancel_left ];
  by_cases h : Real.sin ( ∠ A P B ) = 0 <;> by_cases
h' : Real.sin ( ∠ A P C ) = 0 <;> simp_all +decide [
division_def ];
  · rw [ Real.sin_eq_zero_iff_of_lt_of_lt ] at
h_sin_rule_AB h_sin_rule_AC;
    · rw [ EuclideanGeometry.angle, ] at ⋆ ; simp_all
+decide [ dist_comm ] ;
      rw [ InnerProductGeometry.angle_eq_zero_iff ] at
 ⋆ ; simp_all +decide [ sub_eq_iff_eq_add ] ; (
      obtain ⟨ r, hr, rfl ⟩ := h_sin_rule_AB.2; obtain
 ⟨ s, hs, hs' ⟩ := h_sin_rule_AC.2; simp_all +decide [
 sub_eq_iff_eq_add ];
      rw [ affineIndependent_iff_not_collinear ] at
h_affine ; simp_all +decide [ collinear_pair ];
      contrapose! h_affine; simp_all +decide [
collinear_iff_exists_forall_eq_smul_vadd ] ;
      use A, C - A; simp_all +decide [
sub_eq_iff_eq_add ] ;
      exact ⟨ ⟨ 1, by norm_num ⟩, ⟨ s / r, by rw [
div_eq_inv_mul, MulAction.mul_smul, ← hs', smul_smul,
 inv_mul_cancel₀ hr.ne', one_smul, sub_add_cancel ] ⟩
));
    · linarith [ Real.pi_pos, EuclideanGeometry.
angle_nonneg C A P ];
    · refine' lt_of_le_of_ne ( EuclideanGeometry.
angle_le_pi _ _ _ ) _;
      intro H; simp_all +decide [ EuclideanGeometry.
angle ] ;
      rw [ InnerProductGeometry.angle_eq_pi_iff ] at H
;
      obtain ⟨ r, hr, hr' ⟩ := H.2; simp_all +decide [
 sub_eq_iff_eq_add ] ;
      rw [ segment_eq_image ] at h_P_on_BC ; obtain ⟨
s, hs, hs' ⟩ := h_P_on_BC ; simp_all +decide [
sub_eq_iff_eq_add ];
      have h_affine_dep : ∃ (a b c : ℝ), a + b + c =
0 ∧ a · A + b · B + c · C = 0 ∧ (a ≠ 0 ∨ b ≠ 0 ∨ c ≠
 0) := by
```

```
      use (r - 1), (1 - s), (s - r);
      exact ⟨ by ring, by convert sub_eq_zero.mpr hs
' using 1 ; ext ; norm_num ; ring, by contrapose! hr ;
 linarith ⟩;
      obtain ⟨ a, b, c, h₁, h₂, h₃ ⟩ := h_affine_dep;
specialize h_affine { 0, 1, 2 } ( fun i => if i = 0
then a else if i = 1 then b else c ) ; simp_all +
decide [ Finset.sum_range_succ ] ;
      exact h₃.elim ( fun ha => ha <| h_affine ( by
linarith ) ( by simpa only [ add_assoc ] using h₂ )
|>.1 fun hb => hb.elim ( fun hb => hb <| h_affine (
by linarith ) ( by simpa only [ add_assoc ] using h₂ )
 |>.2.1 ) fun hc => hc <| h_affine ( by linarith ) (
by simpa only [ add_assoc ] using h₂ ) |>.2.2;
    · linarith [ Real.pi_pos, EuclideanGeometry.
angle_nonneg B A P ];
    · refine' lt_of_le_of_ne ( EuclideanGeometry.
angle_le_pi _ _ _ ) _;
      rw [ Ne.eq_def, EuclideanGeometry.
angle_eq_pi_iff_sbtw ];
      rintro ⟨ h₁, h₂, h₃ ⟩;
      rw [ affineIndependent_iff_not_collinear ] at
h_affine;
      rw [ collinear_iff_exists_forall_eq_smul_vadd ]
at h_affine;
      refine' h_affine ⟨ B, A -ᵥ B, _ ⟩;
      simp_all +decide [ Wbtw ];
      rw [ affineSegment_eq_segment ] at h₁;
      rw [ segment_eq_image ] at h₁;
      obtain ⟨ θ, hθ, rfl ⟩ := h₁;
      refine' ⟨ _, _ ⟩;
      · rw [ segment_eq_image ] at h_P_on_BC;
        obtain ⟨ θ', hθ', rfl ⟩ := h_P_on_BC;
        use 1 / ( θ · θ' );
        ext ; norm_num ; ring;
        by_cases hθ : θ = 0 <;> by_cases hθ' : θ' = 0
<;> simp_all +decide;
      · exact ⟨ 1, by norm_num ⟩;
  · simp_all +decide [ ← div_eq_mul_inv ];
    rw [ div_eq_div_iff ] at ⋆;
    any_goals intro H; simp_all +decide [ division_def
 ];
    grind

/-
If the ratio of sin(2a) to sin(a) is sqrt(3), then cos
(a) is sqrt(3)/2.
-/
lemma hmmt_206_cos_val
    (α : ℝ)
    (h_ratio : Real.sqrt 3 / 1 = Real.sin (2 * α) /
Real.sin α)
    (h_sin_ne_zero : Real.sin α ≠ 0) :
    Real.cos α = Real.sqrt 3 / 2 := by
  rw [ Real.sin_two_mul, div_eq_div_iff ] at ⋆ <;>
cases lt_or_gt_of_ne h_sin_ne_zero <;> nlinarith [
Real.sin_sq_add_cos_sq α, Real.sqrt_nonneg 3, Real.
sq_sqrt zero_le_three ] ;

end AristotleLemmas

theorem isosceles_triangle_with_point
    (A B C P : Plane)
    (h_affine : AffineIndependent ℝ ![A, B, C])
    (h_isosceles : dist A B = dist A C)
    (h_P_on_BC : P ∈ segment ℝ B C)
    (h_angle : angle B A P = 2 * angle C A P)
    (h_BP : dist B P = Real.sqrt 3)
    (h_CP : dist C P = 1) :
    dist A P = Real.sqrt 2 := by
  -- Use `hmmt_206_sine_rule_ratio` to show `dist B P
/ dist C P = sin(angle B A P) / sin(angle C A P)`.
  have h_sine_ratio : dist B P / dist C P = Real.sin (
angle B A P) / Real.sin (angle C A P) := by
    convert hmmt_206_sine_rule_ratio A B C P
h_isosceles h_P_on_BC _ _ h_affine using 1 <;>
norm_num [ h_BP, h_CP ];
  -- Substitute `dist B P = sqrt 3` and `dist C P = 1`
 and `angle B A P = 2 * angle C A P` into `
h_sine_ratio`.
  rw [h_BP, h_CP, h_angle] at h_sine_ratio;
```

```
  -- Use `hmmt_206_cos_val` to show `cos(angle C A P)
= sqrt 3 / 2`.
  have h_cos_C_A_P : Real.cos (angle C A P) = Real.
sqrt 3 / 2 := by
    rw [ Real.sin_two_mul ] at h_sine_ratio;
    grind;
  -- Use `hmmt_206_algebraic_sol` to show `dist A P =
sqrt 2`.
  have h_dist_A_P : dist A P ^ 2 + dist A C ^ 2 - 2 *
dist A P * dist A C * Real.cos (angle C A P) = 1 ^ 2
∧ dist A P ^ 2 + dist A C ^ 2 - 2 * dist A P * dist A
 C * Real.cos (angle B A P) = (Real.sqrt 3) ^ 2 := by
    have h_dist_A_P : dist A P ^ 2 + dist A C ^ 2 - 2
* dist A P * dist A C * Real.cos (angle C A P) = dist
C P ^ 2 ∧ dist A P ^ 2 + dist A B ^ 2 - 2 * dist A P
* dist A B * Real.cos (angle B A P) = dist B P ^ 2 :=
by
      have h_dist_A_P : ∀ (X Y Z : Plane), dist X Y ^
 2 + dist X Z ^ 2 - 2 * dist X Y * dist X Z * Real.cos
 (angle Y X Z) = dist Y Z ^ 2 := by
        intros X Y Z; rw [ EuclideanGeometry.angle,
dist_eq_norm_vsub, dist_eq_norm_vsub,
dist_eq_norm_vsub ] ; ring;
        rw [ show Y -ᵥ Z = ( Y -ᵥ X ) - ( Z -ᵥ X ) by
 ext ; simp +decide, norm_sub_sq_real ] ; ring;
      rw [ InnerProductGeometry.cos_angle ] ; ring;
      by_cases h : ‖Y -ᵥ X‖ = 0 <;> by_cases h' : ‖Z
-ᵥ X‖ = 0 <;> simp_all +decide [ mul_assoc, mul_comm,
mul_left_comm ];
        · simp_all +decide [ sub_eq_zero ];
        · simp_all +decide [ sub_eq_zero ];
          rw [ norm_sub_rev ];
        · simp_all +decide [ sub_eq_zero ];
          rw [ norm_sub_rev ];
        · simp +decide [ norm_sub_rev, mul_assoc,
mul_comm, mul_left_comm, h, h' ];
          field_simp;
          rw [ mul_assoc, mul_div_cancel_right₀ _ (
mul_ne_zero ( norm_ne_zero_iff.mpr <| sub_ne_zero.mpr
<| by aesop ) ( norm_ne_zero_iff.mpr <| sub_ne_zero.
mpr <| by aesop ) ) ];
      grind;
    grind;
  -- Use `hmmt_206_algebraic_sol` to show `dist A P =
sqrt 2`. Substitute `dist A C = dist A B` into the
equations.
  have h_dist_A_P_eq : dist A P ^ 2 + dist A B ^ 2 -
dist A P * dist A B * Real.sqrt 3 = 1 ∧ dist A P ^ 2
+ dist A B ^ 2 - dist A P * dist A B = 3 := by
    convert h_dist_A_P using 2 <;> norm_num [
h_isosceles, h_cos_C_A_P ] ; ring;
    rw [ h_angle, Real.cos_two_mul ] ; rw [
h_cos_C_A_P ] ; ring;
    norm_num ; ring;
  -- Use `hmmt_206_algebraic_sol` to show `dist A P =
sqrt 2`. Substitute `dist A B = dist A C` into the
equations.
  have h_dist_A_P_eq' : dist A P ^ 2 + dist A B ^ 2 -
dist A P * dist A B * Real.sqrt 3 = 1 ∧ dist A P ^ 2
+ dist A B ^ 2 - dist A P * dist A B = 3 ∧ dist A B ^
 2 - dist A P ^ 2 = Real.sqrt 3 := by
    have h_cos_sum : Real.cos (angle A P B) + Real.cos
 (angle A P C) = 0 := by
      rw [ EuclideanGeometry.angle, EuclideanGeometry.
angle ];
      -- Since $P$ is on the segment $BC$, we have $B
- P = k(C - P)$ for some $k < 0$.
      obtain ⟨k, hk⟩ : ∃ k : ℝ, k < 0 ∧ B -ᵥ P = k · (
C -ᵥ P) := by
        rw [ segment_eq_image ] at h_P_on_BC;
        rcases h_P_on_BC with ⟨ θ, ⟨ hθ₀, hθ₁ ⟩, rfl ⟩;
        refine' ⟨ -θ / ( 1 - θ ), _, _ ⟩;
        · field_simp;
          rw [ neg_lt_zero, div_pos_iff ];
          cases lt_or_eq_of_le hθ₀ <;> cases
lt_or_eq_of_le hθ₁ <;> first | exact Or.inl ⟨ by
linarith, by linarith ⟩ | skip;
            · norm_num [ ⟨θ = 1⟩ ] at *;
            · norm_num [ ← ⟨0 = θ⟩ ] at *;
              exact absurd h_BP ( by positivity );
          · ext ; norm_num ; ring;
```

```
        by_cases h : 1 - θ = 0 <;> simp_all +decide
[ sq, mul_assoc, mul_comm, mul_left_comm ];
          · norm_num [ show θ = 1 by linarith ] at *;
          · field_simp [h]
            ring;
        rw [ hk.2, InnerProductGeometry.
angle_smul_right_of_neg ] <;> norm_num [ hk.1 ];
        rw [ show C - P = - ( P - C ) by abel1,
InnerProductGeometry.angle_neg_right ] ; norm_num;
    have h_cos_sum : Real.cos (angle A P B) = (dist A
P ^ 2 + dist B P ^ 2 - dist A B ^ 2) / (2 * dist A P *
 dist B P) ∧ Real.cos (angle A P C) = (dist A P ^ 2 +
dist C P ^ 2 - dist A C ^ 2) / (2 * dist A P * dist C
 P) := by
      constructor <;> rw [ EuclideanGeometry.angle,
dist_eq_norm_vsub, dist_eq_norm_vsub,
dist_eq_norm_vsub ];
        · rw [ InnerProductGeometry.cos_angle ];
          rw [ show A -ᵥ B = ( A -ᵥ P ) - ( B -ᵥ P ) by
 simp +decide [ sub_eq_add_neg, add_assoc ],
norm_sub_sq_real ] ; ring;
        · rw [ InnerProductGeometry.cos_angle ];
          rw [ show A -ᵥ C = ( A -ᵥ P ) - ( C -ᵥ P ) by
 rw [ vsub_sub_vsub_cancel_right ] ] ; rw [
@norm_sub_sq ℝ ] ; ring;
        exact?;
    by_cases h : Dist.dist A P = 0 <;> simp_all +
decide [ mul_assoc, mul_comm, mul_left_comm ];
    rw [ div_add_div, div_eq_iff ] at * <;> norm_num [
 h, h_sine_ratio.symm ] at *;
    grind;
  rw [ ← sq_eq_sq₀ ] <;> norm_num ; nlinarith [ Real.
sqrt_nonneg 3, Real.sq_sqrt zero_le_three ]
```

**HMMT_2_195.** Given right triangle $ABC$ with $AB = 4$, $BC = 3$, and $CA = 5$. Circle $\omega$ passes through $A$ and is tangent to $BC$ at $C$. Find the radius of $\omega$.

```
theorem circle_radius_in_right_triangle (A B C O :
Plane) (r : ℝ)
    (h_affine_indep : AffineIndependent ℝ ![A, B, C])
    (h_right_angle : angle A B C = Real.pi / 2)
    (h_AB : dist A B = 4)
    (h_BC : dist B C = 3)
    (h_CA : dist C A = 5)
    (h_r_pos : r > 0)
    (h_A_on_ω : A ∈ sphere O r)
    (h_tangent : dist O (orthogonalProjection (
affineSpan ℝ (Set.range ![B, C])) O) = r ∧
                 orthogonalProjection (affineSpan ℝ (
Set.range ![B, C])) O = C) :
    r = 25 / 8
```

```
-- HMMT_2_195
noncomputable section AristotleLemmas

/-
In the 2D plane, if u and v are non-zero orthogonal
vectors, and w is orthogonal to u, then w is parallel
to v.
-/
lemma vector_parallel_of_perp_of_perp_2d (u v w :
Plane) (hu : u ≠ 0) (hv : v ≠ 0)
    (huv : ⟪u, v⟫ = 0) (hw : ⟪w, u⟫ = 0) : ∃ k : ℝ,
w = k · v := by
  -- Since u and v are orthogonal, we can express w as
 a linear combination of u and v.
  obtain ⟨k₁, k₂, hk⟩ : ∃ k₁ k₂ : ℝ, w = k₁ · u + k₂ ·
v := by
    -- Since $u$ and $v$ are orthogonal and non-zero,
they form a basis for the plane.
    have h_basis : ∀ (w : Plane), ∃ (a b : ℝ), w = a ·
u + b · v := by
      -- Since $u$ and $v$ are orthogonal and non-zero
, they form a basis for the 2D plane. Thus, any vector
$w$ can be written as a linear combination of $u$ and
$v$.
```

```
      have h_basis : LinearIndependent ℝ ![u, v] :=
by
        refine' linearIndependent_fin2.2 _;
        simp_all +decide [ inner_smul_left ];
        intro a ha; have := congr_arg ( fun x => ⟪x,
v⟫ ) ha; norm_num [ inner_smul_left, huv ] at this;
aesop;
      have h_span : Submodule.span ℝ (Set.range ![u,
v]) = ⊤ := by
        refine' Submodule.eq_top_of_finrank_eq _;
        rw [ @finrank_span_eq_card ] <;> aesop;
      intro w; replace h_span := Submodule.
mem_span_range_iff_exists_fun ℝ |>.1 ( h_span.symm ▸
Submodule.mem_top : w ∈ Submodule.span ℝ ( Set.range
![ u, v ] ) ) ; aesop;
    exact h_basis w;
  simp_all +decide [ inner_add_left, inner_smul_left
];
  simp_all +decide [ real_inner_comm ];
  use k₂

end AristotleLemmas

theorem circle_radius_in_right_triangle (A B C O :
Plane) (r : ℝ)
    (h_affine_indep : AffineIndependent ℝ ![A, B, C])
    (h_right_angle : angle A B C = Real.pi / 2)
    (h_AB : dist A B = 4)
    (h_BC : dist B C = 3)
    (h_CA : dist C A = 5)
    (h_r_pos : r > 0)
    (h_A_on_ω : A ∈ sphere O r)
    (h_tangent : dist O (orthogonalProjection (
affineSpan ℝ (Set.range ![B, C])) O) = r ∧
                   orthogonalProjection (affineSpan ℝ (
Set.range ![B, C])) O = C) :
    r = 25 / 8 := by
  -- Let $u = C - B$ and $v = A - B$. Since $\angle
ABC = \pi/2$, $u \perp v$. Also $\|u\|=3 \neq 0$ and $
\|v\|=4 \neq 0$.
  set u : Plane := C - B
  set v : Plane := A - B
  have hu : ‖u‖ = 3 := by
    simp_all +decide [ dist_eq_norm', EuclideanSpace.
norm_eq ];
    exact h_BC
  have hv : ‖v‖ = 4 := by
    convert h_AB using 1
  have huv : ⟪u, v⟫ = 0 := by
    rw [ EuclideanGeometry.angle, ] at h_right_angle;
    rw [ InnerProductGeometry.angle, ] at
h_right_angle;
    simp +zetaDelta at *;
    rw [ real_inner_comm, h_right_angle.resolve_right
( by rintro ( h | h ) <;> simp_all +decide [
sub_eq_iff_eq_add ] ) ]
  have hu_ne_zero : u ≠ 0 := by
    aesop_cat
  have hv_ne_zero : v ≠ 0 := by
    exact sub_ne_zero_of_ne <| by aesop_cat;
  -- Since $C$ is the orthogonal projection of $O$
onto the line $BC$, the vector $w = O - C$ is
orthogonal to the line $BC$, so $w \perp u$.
  set w : Plane := O - C
  have hw : ⟪w, u⟫ = 0 := by
    have hw : ⟪O - (EuclideanGeometry.
orthogonalProjection (affineSpan ℝ (Set.range ![B, C])
) O), u⟫ = 0 := by
      refine' inner_eq_zero_symm.mp _;
      convert EuclideanGeometry.
orthogonalProjection_vsub_mem_direction_orthogonal (
affineSpan ℝ ( Set.range ![ B, C ] ) ) O _ using 1;
      swap;
      exact C - B;
      simp +decide [ direction_affineSpan,
inner_sub_left, inner_sub_right ];
      simp +decide [ vectorSpan_pair,
sub_eq_iff_eq_add ];
      simp +zetaDelta at *;
      norm_num [ inner_sub_left, inner_sub_right ] ;
ring;
```

```
      constructor <;> intro <;> linarith;
    grind;
  -- By the helper lemma `
vector_parallel_of_perp_of_perp_2d`, since $u \perp v$
 and $w \perp v$, we have $w \parallel v$. Thus $O - C
 = k(A - B)$ for some $k \in \mathbb{R}$.
  obtain ⟨k, hk⟩ : ∃ k : ℝ, w = k · v := by
    apply vector_parallel_of_perp_of_perp_2d u v w
hu_ne_zero hv_ne_zero huv hw;
  -- We are given $dist(O, C) = r$, so $\|O - C\| = r$
. Thus $\|k(A - B)\| = r \implies |k| \cdot 4 = r \
implies |k| = r/4$.
  have hk_abs : |k| = r / 4 := by
    have hk_abs : ‖w‖ = r := by
      simp +zetaDelta at *;
      simpa [ h_tangent.2 ] using h_tangent.1;
    rw [ ← hk_abs, hk, norm_smul, hv ] ; ring ; aesop
;
  -- We have $O - A = (O - C) + (C - A) = k(A - B) + (
C - B) - (A - B) = (k-1)(A - B) + (C - B)$.
  have hOA : O - A = (k - 1) · v + u := by
    simp +zetaDelta at *;
    exact Eq.symm ( by ext x; have := congr_fun hk x;
norm_num at *; linarith );
  -- Since $A - B \perp C - B$, by Pythagoras: $\|O -
A\|^2 = \|(k-1)(A - B)\|^2 + \|C - B\|^2$.
  have hOA_sq : ‖O - A‖^2 = (k - 1)^2 * ‖v‖^2 + ‖u‖^2
:= by
    rw [ hOA, @norm_add_sq ℝ ];
    simp_all +decide [ norm_smul, inner_smul_left,
inner_smul_right ];
    cases hw <;> simp_all +decide [ mul_pow, abs_mul
];
    linarith;
  -- We are given $A \in sphere(O, r)$, so $\|O - A\|
= r$.
  have hOA_r : ‖O - A‖ = r := by
    simpa [ norm_sub_rev ] using h_A_on_ω;
  cases abs_cases k <;> push_cast [ * ] at * <;>
nlinarith
```

**HMMT_11_794.** $HOW$, $BOW$, and $DAH$ are equilateral triangles with $WO = 7$ and $AH = 2$. Given that $D, A, B$ are collinear in that order, find $BA$.

```
theorem equilateral_triangles_problem
    (H O W B D A : Plane)
    (h_HOW_affine_independent : AffineIndependent ℝ
![H, O, W])
    (h_BOW_affine_independent : AffineIndependent ℝ
![B, O, W])
    (h_DAH_affine_independent : AffineIndependent ℝ
![D, A, H])
    (h_HOW_equilateral : dist H O = dist O W ∧ dist O
W = dist W H ∧ dist W H = dist H O)
    (h_BOW_equilateral : dist B O = dist O W ∧ dist O
W = dist W B ∧ dist W B = dist B O)
    (h_DAH_equilateral : dist D A = dist A H ∧ dist A
H = dist H D ∧ dist H D = dist D A)
    (h_WO_eq_7 : dist W O = 7)
    (h_AH_eq_2 : dist A H = 2)
    (h_DAB_collinear : Collinear ℝ {D, A, B})
    (h_DAB_order : Sbtw ℝ D A B) :
    dist B A = 11
```

```
-- HMMT_11_794
noncomputable section AristotleLemmas

/-
Algebraic solution for the system of equations derived
 from the Law of Cosines.
-/
theorem algebra_helper (x u : ℝ) (hx : 0 < x)
  (h1 : 4 = x^2 + 147 - x * u)
  (h2 : 4 = (x + 2)^2 + 147 - (x + 2) * u) :
  x = 11 := by
  nlinarith only [ h1, h2, hx ]
```

```
/-
If two equilateral triangles share a side and are
distinct, the square of the distance between the other
 two vertices is 3 times the square of the side length
.
-/
theorem
dist_sq_eq_three_mul_sq_of_equilateral_share_side
    (H O W B : Plane) (s : ℝ) (hs : s > 0)
    (h_HOW : dist H O = s ∧ dist O W = s ∧ dist W H =
 s)
    (h_BOW : dist B O = s ∧ dist O W = s ∧ dist W B =
 s)
    (h_ne : H ≠ B) :
    dist H B ^ 2 = 3 * s^2 := by
  simp_all +decide [ dist_comm, dist_eq_norm,
EuclideanSpace.norm_eq ];
  norm_num [ Real.sqrt_eq_iff_mul_self_eq_of_pos hs ]
at *;
  rw [ Real.sq_sqrt ( by positivity ) ];
  by_contra h_contra;
  -- Since $H$ and $B$ are distinct points, the
vectors $\overrightarrow{OH}$ and $\overrightarrow{OB}
$ are not parallel.
  -- Therefore, the determinant of the matrix formed
by these vectors is non-zero.
  have h_det : (H 0 - O 0) * (W 1 - O 1) - (H 1 - O 1)
 * (W 0 - O 0) ≠ 0 := by
    intro h_det_zero;
    by_cases h_cases : W 0 = O 0;
    · simp_all +decide [ sub_eq_iff_eq_add ];
      cases h_det_zero <;> simp_all +decide [
funext_iff, Fin.forall_fin_two ];
      · grind;
      · exact False.elim <| h_ne <| by ext i;
fin_cases i <;> nlinarith! only [ h_HOW, h_BOW, hs ] ;
    · grind;
  -- By contradiction, assume that $H$ and $B$ are not
 symmetric with respect to $OW$.
  by_cases h_symm : (H 0 - B 0) * (W 1 - O 1) - (H 1 -
 B 1) * (W 0 - O 0) = 0;
  · by_cases h_symm : (H 0 - B 0) * (W 0 - O 0) + (H 1
 - B 1) * (W 1 - O 1) = 0;
    · by_cases h_symm : (H 0 - B 0) = 0;
      · simp_all +decide [ sub_eq_iff_eq_add ];
        exact h_ne ( by ext i; fin_cases i <;> aesop )
;
      · grind;
    · exact False.elim <| h_contra <| by nlinarith [
mul_self_pos.mpr h_symm ] ;
  · grind

end AristotleLemmas

theorem equilateral_triangles_problem
    (H O W B D A : Plane)
    (h_HOW_affine_independent : AffineIndependent ℝ
![H, O, W])
    (h_BOW_affine_independent : AffineIndependent ℝ
![B, O, W])
    (h_DAH_affine_independent : AffineIndependent ℝ
![D, A, H])
    (h_HOW_equilateral : dist H O = dist O W ∧ dist O
 W = dist W H ∧ dist W H = dist H O)
    (h_BOW_equilateral : dist B O = dist O W ∧ dist O
 W = dist W B ∧ dist W B = dist B O)
    (h_DAH_equilateral : dist D A = dist A H ∧ dist A
 H = dist H D ∧ dist H D = dist D A)
    (h_WO_eq_7 : dist W O = 7)
    (h_AH_eq_2 : dist A H = 2)
    (h_DAB_collinear : Collinear ℝ {D, A, B})
    (h_DAB_order : Sbtw ℝ D A B) :
    dist B A = 11 := by
  -- Use `
dist_sq_eq_three_mul_sq_of_equilateral_share_side` to
show $HB^2 = 3 * 7^2 = 147$.
  have hHB : dist H B ^ 2 = 3 * 7 ^ 2 := by
    have hHB : dist H B ^ 2 = 3 * (dist O W) ^ 2 := by
      have h_ne : H ≠ B := by
        rintro rfl;
```

```
      have := h_DAH_affine_independent;
      rw [ affineIndependent_iff_not_collinear ] at
this;
      exact this <| by convert h_DAB_collinear using
 1; ext; simp +decide [ Set.mem_range, Set.
mem_insert_iff, Set.mem_singleton_iff ] ; tauto;
    apply
dist_sq_eq_three_mul_sq_of_equilateral_share_side H O
W B (dist O W) (by
      linarith [ dist_pos.mpr ( show O ≠ W by rintro
rfl; norm_num at * ) ]) (by
      grind +ring) (by
      tauto) h_ne;
    rw [ hHB, ← h_WO_eq_7, dist_comm ];
  -- Use the distances $DA = 2$, $AH = 2$, and $AB =
x$ from the problem statement to set up an equation
for $HB^2$ in terms of $x$.
  obtain ⟨x, hx⟩ : ∃ x, dist D A = 2 ∧ dist A H = 2 ∧
 dist A B = x := by
    grind;
  -- Use the distances $DA = 2$, $AH = 2$, and $AB =
x$ from the problem statement to set up an equation
for $HB^2$ in terms of $x$. Use vector arithmetic.
  have hHB_vector : ‖(H -ᵥ D) - ((2 + x) / 2) · (A -ᵥ
D)‖ ^ 2 = dist H B ^ 2 := by
    -- Use the distances $DA = 2$, $AH = 2$, and $AB =
 x$ from the problem statement to set up an equation
for $HB^2$ in terms of $x$. Use vector arithmetic and
the fact that $D$, $A$, and $B$ are collinear.
    have h_collinear : ∃ t : ℝ, B = D + t · (A -ᵥ D)
∧ t > 1 := by
      obtain ⟨ t, ht ⟩ := h_DAB_order;
      obtain ⟨ u, hu ⟩ := t;
      norm_num [ AffineMap.lineMap_apply ] at hu;
      use 1 / u;
      by_cases hu' : u = 0 <;> simp +decide [ hu', ←
hu.2 ] at *;
      exact one_lt_inv₀ ( lt_of_le_of_ne hu.1 ( Ne.
symm <| by rintro rfl; norm_num at * ) ) |>.2 <|
lt_of_le_of_ne hu.2 <| by rintro rfl; norm_num at *;
    obtain ⟨ t, rfl, ht ⟩ := h_collinear; norm_num [
dist_eq_norm, EuclideanSpace.norm_eq ] at *;
    rw [ ← hx.2.2 ];
    rw [ show ( A 0 - ( D 0 + t * ( A 0 - D 0 ) ) ) ^
2 + ( A 1 - ( D 1 + t * ( A 1 - D 1 ) ) ) ^ 2 = ( t -
1 ) ^ 2 * ( ( A 0 - D 0 ) ^ 2 + ( A 1 - D 1 ) ^ 2 ) by
 ring, Real.sqrt_mul ( by positivity ), Real.sqrt_sq (
 by linarith ) ] ; ring;
    grind;
  -- Use the fact that $DA = 2$, $AH = 2$, and $AB =
x$ to simplify the expression for $HB^2$.
  have hHB_simplified : ‖H -ᵥ D‖ ^ 2 + ((2 + x) / 2) ^
 2 * ‖A -ᵥ D‖ ^ 2 - 2 * ((2 + x) / 2) * (H -ᵥ D) ·ᵥ (A
 -ᵥ D) = dist H B ^ 2 := by
    convert hHB_vector using 1 ; norm_num [
EuclideanSpace.norm_eq ] ; ring;
    rw [ Real.sq_sqrt, Real.sq_sqrt, Real.sq_sqrt ]
<;> try nlinarith only [ sq_nonneg ( H 0 - D 0 ),
sq_nonneg ( H 1 - D 1 ), sq_nonneg ( A 0 - D 0 ),
sq_nonneg ( A 1 - D 1 ) ] ;
    nlinarith only [ sq_nonneg ( H 0 - D 0 - ( A 0 - D
 0 ) * ( 2 + x ) / 2 ), sq_nonneg ( H 1 - D 1 - ( A 1
- D 1 ) * ( 2 + x ) / 2 ), sq_nonneg ( A 0 - D 0 ),
sq_nonneg ( A 1 - D 1 ), sq_nonneg x ];
  -- Use the distances $DA = 2$, $AH = 2$, and $AB =
x$ from the problem statement to simplify the
expression for $HB^2$ further.
  have hHB_final : ‖H -ᵥ D‖ ^ 2 = 4 ∧ ‖A -ᵥ D‖ ^ 2 =
4 ∧ (H -ᵥ D) ·ᵥ (A -ᵥ D) = 2 := by
    norm_num [ dist_eq_norm, EuclideanSpace.norm_eq ]
at *;
    grind;
  rw [ dist_comm ] ; nlinarith [ show 0 ≤ x by
linarith [ @dist_nonneg _ _ A B ] ] ;
```

**HMMT 2 605.** Triangle $ABC$ has $AB = 1$, $BC = \sqrt{7}$, and $CA = \sqrt{3}$. Let $\ell_1$ be the line through $A$ perpendicular

to $AB$, $\ell_2$ the line through $B$ perpendicular to $AC$, and $P = \ell_1 \cap \ell_2$. Find $PC$.

```
theorem find_PC
    (A B C P : Plane)
    (h_affine_indep : AffineIndependent ℝ ![A, B, C])
    (hAB : dist A B = 1)
    (hBC : dist B C = Real.sqrt 7)
    (hCA : dist C A = Real.sqrt 3)
    (hℓ₁ : ℓ₁ = AffineSubspace.mk' A (Submodule.
orthogonal (Submodule.span ℝ {B − A})))
    (hℓ₂ : ℓ₂ = AffineSubspace.mk' B (Submodule.
orthogonal (Submodule.span ℝ {C − A})))
    (hP : P ∈ (ℓ₁ ⊓ ℓ₂ : AffineSubspace ℝ Plane)) :
    dist P C = 3
```

```
    -- HMMT_2_605
theorem find_PC
    (A B C P : Plane)
    (h_affine_indep : AffineIndependent ℝ ![A, B, C])
    (hAB : dist A B = 1)
    (hBC : dist B C = sqrt 7)
    (hCA : dist C A = sqrt 3)
    (hℓ₁ : ℓ₁ = AffineSubspace.mk' A (Submodule.
orthogonal (Submodule.span ℝ {B − A})))
    (hℓ₂ : ℓ₂ = AffineSubspace.mk' B (Submodule.
orthogonal (Submodule.span ℝ {C − A})))
    (hP : P ∈ (ℓ₁ ⊓ ℓ₂ : AffineSubspace ℝ Plane)) :
    dist P C = 3 := by
  simp_all +decide [ dist_eq_norm, EuclideanSpace.
norm_eq ];
  -- By definition of perpendicular bisectors, we know
  that $PA = PB$ and $PB = PC$.
  have hPA_PB : (P 0 − A 0) * (B 0 − A 0) + (P 1 − A
1) * (B 1 − A 1) = 0 := by
    -- Since $P$ lies on the perpendicular line $\
ell_1$, we have $(P − A) \cdot (B − A) = 0$.
    have hP_ell1 : (P 0 − A 0) * (B 0 − A 0) + (P 1 −
A 1) * (B 1 − A 1) = 0 := by
      have hP_ell1_def : P −ᵥ A ∈ (Submodule.span ℝ {
B −ᵥ A})⊥ := by
        exact hP.1
      convert hP_ell1_def ( B −ᵥ A ) ( Submodule.
mem_span_singleton_self _ ) using 1 ; norm_num;
      norm_num [ EuclideanSpace.
inner_eq_star_dotProduct ] ; ring;
    exact hP_ell1
  have hPB_PC : (P 0 − B 0) * (C 0 − A 0) + (P 1 − B
1) * (C 1 − A 1) = 0 := by
    obtain ⟨ hP₁, hP₂ ⟩ := hP; simp_all +decide [
Submodule.mem_orthogonal' ] ;
    convert hP₂ ( C − A ) ( Submodule.
mem_span_singleton_self _ ) using 1 ; ring;
    norm_num [ Fin.sum_univ_two, inner ] ; ring!;
  rw [ Real.sqrt_eq_iff_mul_self_eq_of_pos ] at * <;>
norm_num at *;
  grind
```

**HMMT_2_1185.** Triangle $ABC$ has $AB = 21$, $BC = 55$, and $CA = 56$. There are two points $P$ in the plane such that $\angle BAP = \angle CAP$ and $\angle BPC = 90°$. Find the distance between the two such points.

```
theorem triangle_angle_bisector_right_angle_property
    (A B C P₁ P₂ : Plane)
    (h_affine_indep : AffineIndependent ℝ ![A, B, C])
    (hAB : dist A B = 21)
    (hBC : dist B C = 55)
    (hCA : dist C A = 56)
    (hP₁_angle : angle B A P₁ = angle C A P₁)
    (hP₂_angle : angle B A P₂ = angle C A P₂)
    (hP₁_right_angle : angle B P₁ C = Real.pi / 2)
    (hP₂_right_angle : angle B P₂ C = Real.pi / 2)
    (hP₁_ne_P₂ : P₁ ≠ P₂) :
    dist P₁ P₂ = (5/2) * Real.sqrt 409
```

```
-- HMMT_2_1185
noncomputable section AristotleLemmas

/-
The condition for a point $A + tu$ to lie on the
circle with diameter $BC$ is given by the quadratic
equation $t^2 − t(u \cdot (B−A+C−A)) + (B−A)\cdot(C−A)
 = 0$.
-/
lemma circle_intersections_quadratic
    (A B C : Plane) (u : Plane) (hu : ‖u‖ = 1) (t : ℝ)
 :
    inner ℝ (A + t · u − B) (A + t · u − C) = 0 ↔
    t^2 − t * inner ℝ u (B − A + C − A) + inner ℝ (B −
A) (C − A) = 0 := by
  norm_num [ add_mul, sub_mul, mul_sub,
inner_add_right, inner_add_left, inner_sub_right,
inner_sub_left, inner_smul_right, inner_smul_left ] ;
ring;
  norm_num [ real_inner_comm, hu, EuclideanSpace.
norm_eq ] ; ring;
  norm_num [ EuclideanSpace.norm_eq,
inner_self_eq_norm_sq_to_K ] at * ; aesop

/-
The distance between two roots $t_1, t_2$ of $t^2 + bt
 + c = 0$ is $\sqrt{b^2 − 4c}$, or they are equal (if
the discriminant is non-positive or if we just picked
the same root twice). Actually, if they are distinct
roots of the quadratic, then $|t_1 − t_2| = \sqrt{\
Delta}$. If $t_1 = t_2$, then the distance is 0. Note
that if $\Delta < 0$, there are no real roots, so the
premises are false, but if they were true in some
extension, this formula holds. In Reals, if roots
exist, $\Delta \ge 0$.
Let's just state: if $t_1, t_2$ are roots of $x^2 + bx
 + c = 0$, then $(t_1 − t_2)^2 = b^2 − 4c$. Wait, that
's only true if they are *the* two roots.
If $t_1, t_2$ are roots, then $t_1 = \frac{−b \pm \
sqrt{\Delta}}{2}$.
So $|t_1 − t_2|$ is either $0$ or $\sqrt{\Delta}$.
Let's state it as: $(t_1 − t_2)^2 = b^2 − 4c$ is NOT
necessarily true if $t_1 = t_2$ but $\Delta > 0$ (
which is impossible if they are *all* the roots, but
here we just say they are *some* roots).
Actually, if $t_1$ and $t_2$ are roots, then $t^2 + bt
 + c = (t − t_1)(t − t_2)$ is only true if $t_1, t_2$
are the *only* roots.
Better: If $t_1 \ne t_2$ and both are roots, then they
 are the two roots, so $|t_1 − t_2| = \sqrt{b^2 − 4c}$
.
-/
lemma roots_distance_formula
    (t₁ t₂ b c : ℝ)
    (h₁ : t₁^2 + b * t₁ + c = 0)
    (h₂ : t₂^2 + b * t₂ + c = 0) :
    |t₁ − t₂| = Real.sqrt (b^2 − 4 * c) ∨ t₁ = t₂ :=
by
  -- If $t_1 \neq t_2$, then by Vieta's formulas, $t_1
 + t_2 = −b$ and $t_1 t_2 = c$.
  by_cases h_eq : t₁ = t₂;
  · exact Or.inr h_eq;
  · rw [ ← sq_eq_sq₀ ] <;> norm_num;
    exact Or.inl ( by rw [ Real.sq_sqrt ] <;>
nlinarith [ mul_self_pos.mpr ( sub_ne_zero.mpr h_eq ),
 sq_nonneg ( t₁ + t₂ + b ) ] )

/-
Given the side lengths, we calculate $\cos A = 23/98$
using the Law of Cosines, and $(B−A)\cdot(C−A) = 276$.
-/
lemma triangle_calculations
    (A B C : Plane)
    (hAB : dist A B = 21)
    (hBC : dist B C = 55)
    (hCA : dist C A = 56) :
    Real.cos (angle B A C) = 23 / 98 ∧
    inner ℝ (B − A) (C − A) = 276 := by
  norm_num [ EuclideanSpace.norm_eq, dist_eq_norm,
EuclideanGeometry.angle ] at *;
```

```
    norm_num [ Real.sqrt_eq_iff_mul_self_eq_of_pos,
InnerProductGeometry.angle ] at *;
    norm_num [ EuclideanSpace.norm_eq, inner_sub_left,
inner_sub_right ] at *;
    norm_num [ Fin.sum_univ_two, inner ] at *;
    rw [ Real.cos_arccos ] <;> norm_num [ ← hAB, ← hBC
, ← hCA ];
    · exact ⟨ by rw [ div_eq_iff ( by exact ne_of_gt (
mul_pos ( Real.sqrt_pos.mpr ( by linarith ) ) ( by
norm_num ) ) ) ] ; nlinarith only [ hAB, hBC, hCA,
Real.sqrt_nonneg ( ( B 0 - A 0 ) ^ 2 + ( B 1 - A 1 ) ^
 2 ), Real.mul_self_sqrt ( by nlinarith : 0 ≤ ( B 0 -
 A 0 ) ^ 2 + ( B 1 - A 1 ) ^ 2 ) ], by linarith ⟩;
    · rw [ le_div_iff₀ ] <;> nlinarith only [ Real.
sqrt_nonneg ( ( B 0 - A 0 ) ^ 2 + ( B 1 - A 1 ) ^ 2 ),
 Real.mul_self_sqrt ( by positivity : 0 ≤ ( B 0 - A 0
 ) ^ 2 + ( B 1 - A 1 ) ^ 2 ), hAB, hBC, hCA ];
    · rw [ div_le_iff₀ ] <;> nlinarith only [ hAB, hBC,
hCA, Real.sqrt_nonneg ( ( B 0 - A 0 ) ^ 2 + ( B 1 - A
1 ) ^ 2 ), Real.mul_self_sqrt ( by positivity : 0 ≤ (
 B 0 - A 0 ) ^ 2 + ( B 1 - A 1 ) ^ 2 ) ]

/-
Let $v_B$ and $v_C$ be unit vectors along $AB$ and
$AC$. Let $w = v_B + v_C$.
Then $\|w\| = 11/7$ and the dot product of the unit
bisector $u = w/\|w\|$ with $AB+AC$ is $121/2$.
Calculations:
$\|w\|^2 = 1 + 1 + 2 \cos A = 2(1 + 23/98) = 121/49$.
So $\|w\| = 11/7$.
$u \cdot (AB+AC) = \frac{1}{\|w\|} (v_B + v_C) \cdot
(21 v_B + 56 v_C)$
$= \frac{7}{11} (21 + 56 + 77 \cos A) = \frac{7}{11}
(77 + 77(23/98)) = \frac{7}{11} \cdot 77 (1 + 23/98) =
 49 (121/98) = 121/2$.
-/
lemma bisector_vector_calculation
    (A B C : Plane)
    (hAB : dist A B = 21)
    (hBC : dist B C = 55)
    (hCA : dist C A = 56)
    (h_cos : Real.cos (angle B A C) = 23 / 98) :
    let vB := (1 / 21 : ℝ) · (B - A)
    let vC := (1 / 56 : ℝ) · (C - A)
    let w := vB + vC
    ‖w‖ = 11 / 7 ∧
    inner ℝ ((1 / ‖w‖) · w) (B - A + C - A) = 121 / 2
:= by
    -- Calculate the norm of $w$.
    have h_norm_w : ‖((1 / 21) : ℝ) · (B - A) + ((1 /
56) : ℝ) · (C - A)‖ = 11 / 7 := by
        -- Calculate the norm of $w$ using the given
cosine value.
        have hw_norm : ‖(1 / 21 : ℝ) · (B - A) + (1 / 56 :
ℝ) · (C - A)‖^2 = (11 / 7)^2 := by
            norm_num [ EuclideanSpace.norm_eq, dist_eq_norm'
 ] at *;
            rw [ Real.sq_sqrt ( by positivity ) ] ; rw [
Real.sqrt_eq_iff_mul_self_eq_of_pos ] at * <;>
norm_num at * ; linarith;
        rw [ ← sq_eq_sq₀ ( norm_nonneg _ ) ( by norm_num
), hw_norm ];
    simp_all +decide [ EuclideanSpace.norm_eq, Fin.
sum_univ_two ];
    norm_num [ EuclideanSpace.norm_eq, dist_eq_norm,
EuclideanSpace.norm_eq, Fin.sum_univ_two ] at *;
    norm_num [ EuclideanSpace.norm_eq, Real.
sqrt_eq_iff_mul_self_eq_of_pos, Fin.sum_univ_two,
inner ] at *;
    linarith

/-
If $\angle B A P = \angle C A P$, then $P$ lies on the
 line passing through $A$ with direction $w = v_B +
v_C$, where $v_B, v_C$ are unit vectors along $AB, AC$
.
Proof:
1. $v_B, v_C$ are unit vectors.
2. $w = v_B + v_C$ is the internal bisector direction.
3. The condition $\angle B A P = \angle C A P$ implies
 $P$ lies on the internal or external bisector.
```

```
4. As analyzed, for `EuclideanGeometry.angle` (which
is in $[0, \pi]$), the condition $\angle(AB, AP) = \
angle(AC, AP)$ is satisfied by both the internal
bisector (angle $\alpha/2$) and the opposite ray (
angle $\pi - \alpha/2$).
5. The external bisector (perpendicular to internal)
has angles $(\pi+\alpha)/2$ and $(\pi-\alpha)/2$,
which are not equal unless $\alpha=0$.
6. Thus $P$ must lie on the line spanned by $w$.
7. So $P - A$ is collinear with $w$.
8. Thus $P = A + t w$.
-/
lemma angle_bisector_locus
    (A B C P : Plane)
    (hAB : dist A B = 21)
    (hBC : dist B C = 55)
    (hCA : dist C A = 56)
    (h_cos : Real.cos (angle B A C) = 23 / 98)
    (hP : angle B A P = angle C A P)
    (hP_ne_A : P ≠ A) :
    let vB := (1 / 21 : ℝ) · (B - A)
    let vC := (1 / 56 : ℝ) · (C - A)
    let w := vB + vC
    ∃ t : ℝ, P = A + t · w := by
    simp_all +decide [ EuclideanGeometry.angle,
dist_eq_norm', norm_smul, norm_div ];
    -- By definition of angle bisector, we know that $P$
 lies on the line passing through $A$ with direction
$w = v_B + v_C$, where $v_B, v_C$ are unit vectors
along $AB, AC$.
    obtain ⟨t₁, t₂, ht⟩ : ∃ t₁ t₂ : ℝ, P - A = t₁ · (B -
 A) + t₂ · (C - A) := by
        -- Since $B - A$ and $C - A$ are linearly
independent, we can express $P - A$ as a linear
combination of these vectors.
        have h_lin_indep : LinearIndependent ℝ ![B - A, C
 - A] := by
            refine' linearIndependent_fin2.mpr _;
            refine' ⟨ sub_ne_zero.mpr _, fun a ha => _ ⟩;
            · aesop_cat;
            · simp_all +decide [ norm_sub_rev ];
                rw [ show B = A + a · ( C - A ) by rw [ ha,
add_sub_cancel ] ] at hAB hBC ; norm_num [ norm_smul,
hCA ] at hAB hBC;
                rw [ norm_sub_rev ] at *;
                rw [ norm_sub_rev ] at hAB ; norm_num [ hCA ]
at hAB;
                rw [ show C - ( A + a · ( C - A ) ) = ( 1 - a
 ) · ( C - A ) by ext ; simpa using by ring, norm_smul,
 Real.norm_eq_abs ] at hBC ; norm_num [ hCA ] at hBC;
                cases abs_cases a <;> cases abs_cases ( 1 - a
 ) <;> linarith;
        have h_span : Submodule.span ℝ (Set.range ![B - A,
 C - A]) = ⊤ := by
            refine' Submodule.eq_top_of_finrank_eq _;
            rw [ finrank_span_eq_card ] <;> aesop;
        replace h_span := SetLike.ext_iff.mp h_span ( P -
A ) ; simp_all +decide [ Submodule.mem_span_insert,
Submodule.mem_span_singleton ];
        obtain ⟨ a, b, h ⟩ := h_span; exact ⟨ b, a, by rw
[ h, add_comm ] ⟩ ;
    -- Since $\angle BAP = \angle CAP$, we have $\cos(\
angle BAP) = \cos(\angle CAP)$.
    have h_cos_eq : inner ℝ (B - A) (t₁ · (B - A) + t₂ ·
 (C - A)) * ‖C - A‖ = inner ℝ (C - A) (t₁ · (B - A) +
t₂ · (C - A)) * ‖B - A‖ := by
        have h_cos_eq : Real.cos (InnerProductGeometry.
angle (B - A) (t₁ · (B - A) + t₂ · (C - A))) = Real.cos
 (InnerProductGeometry.angle (C - A) (t₁ · (B - A) + t
₂ · (C - A))) := by
            grind;
        rw [ InnerProductGeometry.cos_angle,
InnerProductGeometry.cos_angle ] at h_cos_eq;
        rw [ div_eq_div_iff ] at h_cos_eq <;> norm_num at
*;
        · exact mul_left_cancel₀ ( show ‖t₁ · ( B - A ) + t
₂ · ( C - A )‖ ≠ 0 from norm_ne_zero_iff.mpr <| by
intro h; simp_all +decide [ sub_eq_iff_eq_add ] ) <|
by linear_combination' h_cos_eq;
        · exact ⟨ sub_ne_zero_of_ne <| by rintro rfl;
norm_num at hAB, by rw [ ← ht ] ; exact
```

```
  sub_ne_zero_of_ne hP_ne_A );
    · exact ⟨ sub_ne_zero_of_ne <| by aesop_cat, by
intro h; simp_all +decide [ sub_eq_iff_eq_add ] );
  simp_all +decide [ norm_sub_rev, inner_add_left,
inner_add_right, inner_smul_left, inner_smul_right ];
  simp_all +decide [ norm_eq_sqrt_real_inner,
real_inner_comm ];
  simp_all +decide [ Real.
sqrt_eq_iff_mul_self_eq_of_pos,
inner_self_eq_norm_sq_to_K ];
  -- By simplifying, we can see that $t₁ = t₂ * (56 /
21)$.
  have h_t_eq : t₁ = t₂ * (56 / 21) := by
    rw [ InnerProductGeometry.cos_angle ] at h_cos;
    rw [ div_eq_iff ] at h_cos <;> norm_num [
norm_sub_rev ] at *;
    · norm_num [ hAB, hCA, h_cos ] at h_cos_eq ;
linarith;
    · exact ⟨ sub_ne_zero_of_ne <| by rintro rfl;
norm_num at hAB, sub_ne_zero_of_ne <| by rintro rfl;
norm_num at hCA );
    exact ⟨ t₂ * 56, by ext ; norm_num ; have :=
congr_fun ht ⟨_⟩ ; norm_num at * ; rw [ h_t_eq ] at
this ; linarith ⟩

end AristotleLemmas

theorem triangle_angle_bisector_right_angle_property
    (A B C P₁ P₂ : Plane)
    (h_affine_indep : AffineIndependent ℝ ![A, B, C])
    (hAB : dist A B = 21)
    (hBC : dist B C = 55)
    (hCA : dist C A = 56)
    (hP₁_angle : angle B A P₁ = angle C A P₁)
    (hP₂_angle : angle B A P₂ = angle C A P₂)
    (hP₁_right_angle : angle B P₁ C = Real.pi / 2)
    (hP₂_right_angle : angle B P₂ C = Real.pi / 2)
    (hP₁_ne_P₂ : P₁ ≠ P₂) :
    dist P₁ P₂ = (5/2) * Real.sqrt 409 := by
  -- Use `angle_bisector_locus` to show $P_1 = A + t_1
  w$ and $P_2 = A + t_2 w$ for some $t_1, t_2$, where
$w$ is the bisector vector.
  obtain ⟨t₁, ht₁⟩ : ∃ t₁ : ℝ, P₁ = A + t₁ · ((1 / 21 :
ℝ) · (B − A) + (1 / 56 : ℝ) · (C − A)) := by
    have h_cos : Real.cos (angle B A C) = 23 / 98 :=
by
      convert ( triangle_calculations A B C hAB hBC
hCA ) |> And.left using 1;
    apply angle_bisector_locus A B C P₁ hAB hBC hCA
h_cos hP₁_angle (by
      rintro rfl; norm_num at *;
      norm_num [ hP₁_right_angle ] at h_cos)
  obtain ⟨t₂, ht₂⟩ : ∃ t₂ : ℝ, P₂ = A + t₂ · ((1 / 21 :
ℝ) · (B − A) + (1 / 56 : ℝ) · (C − A)) := by
    apply_rules [angle_bisector_locus ];
    · convert ( triangle_calculations A B C hAB hBC
hCA ) |> And.left using 1;
    · rintro rfl; norm_num at *;
      simp_all +decide [ EuclideanGeometry.angle,
dist_eq_norm ];
      have := InnerProductGeometry.cos_angle ( B − P₂
) ( C − P₂ ) ; simp_all +decide [ norm_sub_rev ] ;
      have := norm_sub_sq_real ( B − P₂ ) ( C − P₂ ) ;
norm_num [ hAB, hCA, hBC ] at this;
      linarith;
  -- Let $u = \frac{1}{\|w\|} w$. Then $u$ is a unit
vector.
  set u : Plane := (1 / ∥(1 / 21 : ℝ) · (B − A) + (1 /
56 : ℝ) · (C − A)∥) · ((1 / 21 : ℝ) · (B − A) + (1 /
56 : ℝ) · (C − A)) with hu_def
  have hu_unit : ∥u∥ = 1 := by
    rw [ norm_smul, Real.norm_of_nonneg ( by
positivity ), div_mul_cancel₀ _ ( ne_of_gt <|
norm_pos_iff.mpr _ ) ];
    intro h; simp_all +decide [ sub_eq_iff_eq_add ] ;
    rw [ ← eq_sub_iff_add_eq' ] at h ; norm_num [
sub_eq_iff_eq_add ] at h ; aesop;
  have hu_dot : inner ℝ u (B − A + C − A) = 121 / 2
:= by
    have := bisector_vector_calculation A B C hAB hBC
hCA ( triangle_calculations A B C hAB hBC hCA |>.1 ) ;
```

```
  aesop;
  have hP₁_P₂ : ∃ s₁ s₂ : ℝ, P₁ = A + s₁ · u ∧ P₂ = A
+ s₂ · u ∧ s₁^2 − s₁ * (121 / 2) + 276 = 0 ∧ s₂^2 − s₂
* (121 / 2) + 276 = 0 ∧ s₁ ≠ s₂ := by
    -- Use `circle_intersections_quadratic` with
vector $u$ and scalar $s_i$.
    have hP₁_quad : t₁^2 * ∥(1 / 21 : ℝ) · (B − A) +
(1 / 56 : ℝ) · (C − A)∥^2 − t₁ * ∥(1 / 21 : ℝ) · (B −
A) + (1 / 56 : ℝ) · (C − A)∥ * (121 / 2) + 276 = 0 :=
by
      have hP₁_quad : inner ℝ (A + t₁ · ((1 / 21 : ℝ)
· (B − A) + (1 / 56 : ℝ) · (C − A)) − B) (A + t₁ · ((1
/ 21 : ℝ) · (B − A) + (1 / 56 : ℝ) · (C − A)) − C) = 0
:= by
        rw [ ← ht₁ ];
        -- By definition of angle, $\angle B P_1 C = \
pi / 2$ implies that $(P_1 − B) \cdot (P_1 − C) = 0$.
        have hP₁_dot : inner ℝ (P₁ − B) (P₁ − C) = 0
:= by
          have h_angle : EuclideanGeometry.angle B P₁
C = Real.pi / 2 := hP₁_right_angle
          rw [ EuclideanGeometry.angle ] at h_angle;
          rw [ InnerProductGeometry.angle ] at h_angle
;
          norm_num [ sub_eq_iff_eq_add ] at h_angle;
          rcases h_angle with h | rfl | rfl <;>
norm_num [ inner_sub_left, inner_sub_right ] at *;
          linarith;
          exact hP₁_dot;
      have hP₁_quad : t₁^2 * ∥(1 / 21 : ℝ) · (B − A) +
(1 / 56 : ℝ) · (C − A)∥^2 − t₁ * inner ℝ ((1 / 21 :
ℝ) · (B − A) + (1 / 56 : ℝ) · (C − A)) (B − A + C − A)
+ inner ℝ (B − A) (C − A) = 0 := by
        convert hP₁_quad using 1 ; norm_num [
inner_sub_left, inner_sub_right, inner_smul_left,
inner_smul_right ] ; ring;
        norm_num [ norm_add_sq_real, norm_smul,
inner_add_left, inner_add_right, inner_smul_left,
inner_smul_right ] ; ring;
        norm_num [ norm_sub_sq_real, inner_sub_left,
inner_sub_right ] ; ring;
        norm_num [ real_inner_comm,
real_inner_self_eq_norm_sq ] ; ring;
      convert hP₁_quad using 2 ; norm_num [ hu_def,
hu_unit, hu_dot ] ; ring;
      · rw [ ← hu_dot ] ; norm_num [ hu_def, hu_unit,
hu_dot ] ; ring;
        norm_num [ ← smul_add, inner_smul_left,
inner_smul_right ] ; ring;
        by_cases h : ∥( 1 / 21 : ℝ ) · ( B − A ) + ( 1
/ 56 : ℝ ) · ( C − A )∥ = 0 <;> aesop;
      · convert ( triangle_calculations A B C hAB hBC
hCA ) |>.2.symm using 1
    have hP₂_quad : t₂^2 * ∥(1 / 21 : ℝ) · (B − A) +
(1 / 56 : ℝ) · (C − A)∥^2 − t₂ * ∥(1 / 21 : ℝ) · (B −
A) + (1 / 56 : ℝ) · (C − A)∥ * (121 / 2) + 276 = 0 :=
by
      have hP₂_quad : inner ℝ (A + t₂ · ((1 / 21 : ℝ)
· (B − A) + (1 / 56 : ℝ) · (C − A)) − B) (A + t₂ · ((1
/ 21 : ℝ) · (B − A) + (1 / 56 : ℝ) · (C − A)) − C) = 0
:= by
        rw [ ← ht₂ ];
        rw [ EuclideanGeometry.angle ] at hP₂
_right_angle;
        rw [ InnerProductGeometry.angle ] at hP₂
_right_angle;
        rw [ show P₂ − B = − ( B − P₂ ) by abel1, show
P₂ − C = − ( C − P₂ ) by abel1, inner_neg_left,
inner_neg_right ] ; aesop;
      have hP₂_quad : t₂^2 * ∥(1 / 21 : ℝ) · (B − A) +
(1 / 56 : ℝ) · (C − A)∥^2 − t₂ * inner ℝ ((1 / 21 :
ℝ) · (B − A) + (1 / 56 : ℝ) · (C − A)) (B − A + C − A)
+ inner ℝ (B − A) (C − A) = 0 := by
        convert hP₂_quad using 1 ; norm_num [
EuclideanSpace.norm_eq, Fin.sum_univ_two ] ; ring;
        rw [ Real.sq_sqrt ] <;> norm_num [
EuclideanSpace.norm_eq, Fin.sum_univ_two ] ; ring;
        · norm_num [ Real.sq_sqrt ( add_nonneg (
sq_nonneg _ ) ( sq_nonneg _ ) ), inner ] ; ring;
        · norm_num [ dist_eq_norm, EuclideanSpace.
norm_eq ] at *;
```

```
        rw [ Real.sqrt_eq_iff_mul_self_eq_of_pos ]
at * <;> norm_num at * ; nlinarith only [ hAB, hBC,
hCA ];
      convert hP₂_quad using 2 <;> norm_num [
inner_smul_left, inner_smul_right ] ; ring;
      · rw [ ← hu_dot ] ; norm_num [ u,
inner_smul_left, inner_smul_right ] ; ring;
        norm_num [ ← smul_add, inner_smul_left,
inner_smul_right ] ; ring;
        by_cases h : ‖ ( 1 / 21 : ℝ ) · ( B - A ) + (
1 / 56 : ℝ ) · ( C - A )‖ = 0 <;> aesop;
       · convert ( triangle_calculations A B C hAB hBC
hCA ) |>.2.symm using 1;
    refine' ⟨ t₁ * ‖ ( 1 / 21 : ℝ ) · ( B - A ) + ( 1
/ 56 : ℝ ) · ( C - A )‖, t₂ * ‖ ( 1 / 21 : ℝ ) · ( B -
 A ) + ( 1 / 56 : ℝ ) · ( C - A )‖, _, _, _, _, _ ⟩
<;> simp_all +decide [ ← smul_assoc ];
       · by_cases h : ‖ ( 21⁻¹ : ℝ ) · ( B - A ) + ( 56⁻¹
 : ℝ ) · ( C - A )‖ = 0 <;> simp_all +decide [
mul_assoc, mul_comm, mul_left_comm ];
       · by_cases h : ‖ ( 21⁻¹ : ℝ ) · ( B - A ) + ( 56⁻¹
 : ℝ ) · ( C - A )‖ = 0 <;> simp_all +decide [
mul_assoc, mul_comm, mul_left_comm ];
     · linear_combination' hP₁_quad;
     · linear_combination' hP₂_quad;
     · refine' ⟨ _, _ ⟩;
       · exact fun h => hP₁_ne_P₂ <| by rw [ h ] ;
       · intro h; simp_all +decide [ sub_eq_iff_eq_add
] ;
  obtain ⟨ s₁, s₂, rfl, rfl, hs₁, hs₂, hs ⟩ := hP₁_P₂;
 norm_num [ dist_eq_norm, EuclideanSpace.norm_eq ] at
*;
  rw [ Real.sqrt_eq_iff_mul_self_eq_of_pos ] <;>
ring_nf <;> norm_num [ hu_unit ];
  grind
```

