# OpenReview forum: "Euclean: Automated Geometry Problem Formalization with Unified Verification in Lean"
_ICML.cc/2026/Conference — ICML 2026 regular_

### Official Review · Reviewer_fehW · 2026-03-06

**Soundness:** 2
**Presentation:** 3
**Significance:** 3
**Originality:** 3
**Overall Recommendation:** 4
**Confidence:** 3

**Summary:**

This paper presents EUCLEAN, a framework for automatically formalizing natural language geometry problems into LEAN 4 with native Mathlib compatibility. The system uses a four-stage pipeline—constraint explication, configuration anchoring via "prove-first" reasoning, formalization mapping through retrieval, and iterative compiler-feedback repair—to handle the unique challenges of geometry formalization, particularly implicit non-degeneracy conditions and topological constraints. The authors construct two datasets: OMNI-Geometry (768 competition problems) and Numina-Geometry (177,597 problems), claiming them to be the largest Lean geometry formalization datasets. Human evaluation shows 48.89% TOP1 and 73.33% TOP5 accuracy on OMNI-Geometry, and training Goedel v2 on the data improves proof success from 13.6% to 15.1%.

**Compliance With Llm Reviewing Policy:**

Affirmed.

**Final Justification:**

I have no further concerns.

**Key Questions For Authors:**

- Will the EUCLEAN codebase, prompts, and both datasets be publicly released upon acceptance?
- What is the estimated semantic accuracy of Numina-Geometry, given that human evaluation was only conducted on the smaller OMNI-Geometry subset?
- How do you ensure that the 177K compiling formalizations in Numina-Geometry do not contain vacuously true or over-specified statements that trivially compile but misrepresent the original problems?
- Given that the "prove-first" anchoring actually decreases compilation rate (from 52.1% to 50.0%), what evidence supports that the resulting formalizations are semantically superior rather than merely harder to compile?

**Limitations:**

yes

**Strengths And Weaknesses:**

### Strengths

- The choice of Mathlib-native formalization over custom axiom systems is well-motivated, as it enables direct integration with the broader Lean ecosystem and avoids the soundness issues of SMT-based approaches like LeanEuclid.
- The scale of Numina-Geometry (177K problems) represents a significant advance over prior geometry formalization efforts that were limited to hundreds of problems, potentially enabling large-scale training for geometry theorem proving.
- The "prove-first" configuration anchoring strategy is a creative approach to resolving topological ambiguities that naturally arise when translating informal geometry into formal specifications.
- The paper provides concrete, detailed code comparisons (Appendix E) that clearly illustrate the practical differences between EUCLEAN and prior approaches like LeanEuclid and LeanGeo.

### Weaknesses

- The downstream training improvement is marginal (13.6% to 15.1%), and the Aristotle evaluation on only 10 randomly sampled problems is too small to draw meaningful conclusions about the dataset's utility for advancing theorem proving.
- The paper claims the code and datasets will be open-sourced but provides no repository link, anonymous or otherwise, making it impossible for reviewers to verify the claimed scale, quality, or reproducibility of the 177K formalized problems.
- The ablation study is conducted on only 30 problems with compilation rate as the sole metric, which conflates syntactic validity with semantic correctness—a formalization that compiles is not necessarily faithful to the original problem.
- The 48.89% TOP1 semantic accuracy means that more than half of the formalizations in the dataset may not faithfully represent the original problems, yet the paper does not address how this noise affects downstream training or how users should filter the dataset.

---

> ### Author Rebuttal · Authors · 2026-03-31
>
> Thank you for recognizing the motivation of `mathlib`-native geometry formalization, the dataset scale, and the role of configuration anchoring. We understand your main concern as follows: under still-limited semantic evaluation, is this `177K` corpus reliable, reproducible, and practically useful for downstream users?
>
> ### W2 / Q1: release and reproducibility
>
> We will certainly release the full EUCLEAN codebase, all prompts, and the OMNI-Geometry / Numina-Geometry datasets to support reproducibility.
>
> ### W3 / Q3: semantic correctness and how we handle semantic noise
>
> Assessing semantic correctness is intrinsically difficult in geometry for two reasons. First, compared with algebra, natural-language geometry omits more configuration and non-degeneracy information, while native `mathlib` requires these conditions to be explicit in the theorem statement. Second, unlike benchmarks such as `minif2f`, we do not have ground-truth formal statements for automatic symbolic equivalence checking.
>
> Given this difficulty, we filter and validate as much as feasible, relying on layered evidence rather than any single metric: compilation checking guarantees Lean / `mathlib` well-formedness; a preliminary LLM-based semantic screen was used as an auxiliary signal during data construction; OMNI human evaluation reports `48.89%` TOP1 and `73.33%` TOP5; and downstream proving provides an orthogonal utility signal.
>
> Accordingly, our claim is not that the whole dataset is a noise-free gold benchmark. Rather, it is a large-scale, `mathlib`-native geometry formalization corpus with layered validation, substantial semantic signal, and initial downstream utility.
>
> ### W1 / W4 / Q2: estimated Numina quality and downstream use
>
> We agree that `13.6% -> 15.1%` is still modest, and that the original 10-problem Aristotle sample was too small.
>
> We still view proof-trace acquisition as the bottleneck: current provers remain weak on geometry, so traces remain limited. To address this, we expanded the Aristotle experiment to 100 problems. We filtered out cases that Aristotle explicitly judged `False` and proved by contradiction, cases that became trivially easy because of obvious misformalization, and cases for which Aristotle found concrete counterexamples. After this, Aristotle still proves `25` problems, while another `19` remain as `sorry` but are not flagged as problematic and come with no counterexamples. This gives a rough but more informative lower-bound estimate that at least `25%` of Numina is semantically usable, with the true proportion plausibly higher if some of those unresolved-but-unrefuted cases are also valid.
>
> For downstream users, this suggests reranking or filtering the corpus with stronger provers / models, for example by refuting over-strong constraints, constructing counterexamples, or checking whether added conditions are actually necessary.
>
> More generally, large formal corpora are rarely used as fully-gold datasets from the outset. A common workflow is to first obtain a large candidate corpus, then sharpen useful subsets through proving, refutation, reranking, and expert iteration. Corpora such as Lean Workbook likewise show how this strategy supports later provers such as DeepSeek-Prover and Goedel-Prover.
>
> ### Q4: why `prove-first` anchoring lowers compilation, and what it is for
>
> The key point is geometry itself. Compared with algebra, geometry more often leaves configuration and non-degeneracy conditions implicit, while native `mathlib` requires them to be explicit. This is exactly why geometry formalization needs dedicated handling of hidden constraints.
>
> The goal of `prove-first` / configuration anchoring is therefore to reduce under-specified formalizations. Empirically, after adding anchoring, the average number of hypotheses increases from `5.22` to `6.69`, consistent with the intended mechanism: the model makes more previously implicit geometric conditions explicit.
>
> That statistic alone does not prove semantics always improve. If the added constraints are wrong, vacuous/inconsistent, or overly strong, they can hurt both compilation and semantic quality. Still, because this step is necessary in geometry and the observed behavior matches its intended role, it is useful for later filtering and reranking by reducing under-specification and exposing hidden constraints earlier.
>
> ### Summary
>
> Our main points are:
>
> 1. Semantic filtering is difficult in geometry, but we already use layered validation to screen and quantify quality.
> 2. The enlarged Aristotle experiment provides a more credible lower-bound estimate of Numina's semantic quality, and also suggests that downstream users can further filter the corpus with stronger provers / models, which is also a common way large-scale formal corpora are used in practice.
> 3. Given the nature of geometry, `prove-first` is meant to reduce under-specification, and the observed ablation behavior is consistent with that design goal.

---

> > ### Author Rebuttal · Reviewer_fehW · 2026-03-31
> >
> > I have no further questions and will keep my positive score.

---

> > > ### Author Response · Authors · 2026-04-02
> > >
> > > We sincerely thank you for your careful review and for engaging with our rebuttal. We are very glad that our responses helped address your concerns, and we greatly appreciate your positive evaluation. Your feedback has been invaluable in helping us refine and better present our work.

---

### Official Review · Reviewer_JBug · 2026-03-10

**Soundness:** 3
**Presentation:** 3
**Significance:** 3
**Originality:** 2
**Overall Recommendation:** 3
**Confidence:** 2

**Summary:**

The paper presents EUCLEAN, a pipeline for translating informal geometry problems into native Lean and mathlib statements. It combines assumption completion, proof-sketch-based configuration anchoring, concept grounding, and iterative code repair, and uses this pipeline to build large-scale geometry formalization datasets. The paper further shows that these data can be used to obtain modest gains in downstream Lean geometry theorem proving.

**Compliance With Llm Reviewing Policy:**

Affirmed.

**Final Justification:**

I thank the authors, for their detailed and thoughtful rebuttal. The clarifications regarding semantic evaluation, the additional Aristotle analysis, and the discussion of validation methodology are helpful.

However, my main concern remains. The value of the paper depends critically on the semantic correctness of the generated formalizations, and while the authors provide multiple indirect signals of quality, the evidence still suggests that a substantial fraction of the dataset may be noisy. The additional experiments help contextualize this, but do not, in my view, fully resolve the uncertainty around how much of the observed downstream performance is driven by correctly formalized problems.

More broadly, while the pipeline is well-motivated and clearly presented, the current results show only modest downstream gains, and it remains unclear whether the dataset is of sufficient quality to substantially advance theorem proving in practice. Therefore, my assessment remains unchanged.

**Key Questions For Authors:**

- Could the authors clarify how they intend to ensure that the dataset consists of correctly formalized problems from OMNI-Math and Numina-Math?
- Does the score in Table 5 aggregate results over incorrectly formalized problems from Numina-Math? In particular, could it be that a substantial fraction of the roughly 15% correct solutions comes from incorrectly formalized, and potentially easier, problems?

**Limitations:**

yes

**Strengths And Weaknesses:**

Strengths:
- The case study on circle intersections nicely illustrates the difference between LeanEuclid and the authors’ proposed pipeline.
- The presentation is very good. The current gap is clearly articulated, along with a crisp analysis of why this gap arises.
- The ablation study in Section 5.1 is insightful and transparent.
- Figure 1 is helpful and illustrates the method well, although the role of the LLM could be clarified further.

Weaknesses:
- In the caption of Table 3, the authors claim that “each component yields meaningful gains,” which does not appear to be true: adding configuration anchoring leads to a degradation in performance.
- Table 4 indicates that, while the performance is reasonably high, the full generated dataset may only partially consist of faithful formalized statements.
- The performance gain in Table 5 appears rather modest.
- The role of the LLM could be stated more clearly; overall, Section 3.1 is somewhat vague.

Minor:
- Typo in line 126.

My main concern is that the paper’s value depends heavily on the quality of the generated formalizations, yet Table 4 suggests that semantic faithfulness remains limited, while Table 5 shows only modest downstream gains. As a result, it is not yet clear that the proposed pipeline produces data of sufficient quality to substantially improve theorem proving in practice. I would be willing to increase my score if these concerns are addressed.

---

> ### Author Rebuttal · Authors · 2026-03-31
>
> Thank you for the careful review. We understand your main concern: because the paper's value depends heavily on the quality of the generated formalizations, do Tables 4 and 5 really show that the data are reliable and useful?
>
> ### Main concern, Q1, W2: semantic correctness
>
> On semantic quality: establishing semantic equivalence in geometry autoformalization is especially difficult for two reasons.
>
> First, unlike algebra, geometry often omits configuration and non-degeneracy information, such as point ordering, betweenness, sidedness, and concrete topological configuration. In native `mathlib`, these conditions must be written explicitly in the theorem statement.
>
> Second, unlike benchmarks such as `minif2f`, we do not have ground-truth formal statements for automatic symbolic equivalence checking.
>
> Given this difficulty, we filter and validate as much as currently feasible, relying on layered evidence rather than any single metric: compilation checking guarantees Lean / `mathlib` well-formedness; a preliminary LLM-based semantic screen was used as an auxiliary signal during data construction; OMNI human evaluation reports `48.89%` TOP1 and `73.33%` TOP5; and downstream proving provides an orthogonal utility signal.
>
> Taken together, this supports a narrower and more accurate claim: Numina is the first large-scale, `mathlib`-native geometry formalization corpus with layered validation. It contains semantic noise, but already includes a substantial semantically correct subset and shows initial downstream utility.
>
> More generally, large formal corpora are rarely used as fully-gold datasets from the outset. A common workflow is to first obtain a large candidate corpus, then sharpen useful subsets through proving, refutation, reranking, and expert iteration. Corpora such as Lean Workbook likewise show how this strategy supports later provers such as DeepSeek-Prover and Goedel-Prover.
>
> ### Main concern, Q2: data quality, could the solved fraction come from wrong but easier statements?
>
> Under current semantic-evaluation limits, the proving results necessarily mix semantically correct and semantically incorrect formalizations. The more important question is whether the provable part consists mainly of wrongly formalized, trivially easy statements, or whether stronger provers can also recover proofs on a meaningful semantically reasonable subset. We believe the latter is true.
>
> To address this, we expanded the Aristotle experiment to 100 problems. We filtered out cases that Aristotle explicitly judged `False` and proved by contradiction, cases that became trivially easy because of obvious misformalization, and cases for which Aristotle found concrete counterexamples. After this, Aristotle still proves `25` problems, while another `19` remain as `sorry` but are not flagged as problematic and come with no counterexamples. This gives a rough but more informative lower-bound estimate that at least `25%` of Numina is semantically usable, with the true proportion plausibly higher if some of those unresolved-but-unrefuted cases are also valid.
>
> This suggests that provability is not confined to obviously wrong formalizations. Collecting more high-quality traces from stronger provers on the same corpus is thus a promising next step.
>
> ### W1: why configuration anchoring reduces compilation
>
> The caption phrase "each component yields meaningful gains" is imprecise, and we will revise it.
>
> Given the nature of geometry, correct formalization requires making hidden conditions explicit even more than in algebra. Accordingly, the role of `configuration anchoring` is to reduce under-specification rather than simply maximize compile rate. Handling this hidden-constraint problem is exactly why a dedicated pipeline is needed, and why configuration anchoring is one of our core contributions.
>
> ### W4: the role of the LLM and Section 3.1
>
> The model takes the natural-language problem statement as input, analyzes missing constraints through proof reasoning, produces a formalization by mapping to `mathlib` concepts, and then repairs formalization errors based on compiler feedback.
>
> Section 3.1 explains why native `mathlib` geometry formalization needs explicit handling of hidden constraints, whereas Section 3.2 describes the four-stage pipeline.
>
> ### Summary
>
> Our main points are:
>
> 1. Semantic evaluation is intrinsically harder in geometry than in algebra, so we validate quality through layered evidence rather than a single metric; for large formal datasets, the standard practice is often to build a large candidate corpus first and then iteratively filter and strengthen it.
> 2. The enlarged Aristotle experiment indirectly provides evidence about semantic accuracy, and shows that nontrivial, semantically reasonable statements are also provable.
> 3. The geometry-specific need to expose hidden constraints is why Table 3 should be interpreted as above, and why the pipeline design itself is a substantive contribution.

---

> > ### Author Rebuttal · Reviewer_JBug · 2026-04-02
> >
> > Thank you for the rebuttal and clarifications. While helpful, my main concern remains: the paper’s value depends on the quality of the generated formalizations, and the evidence for semantic correctness and downstream impact is still limited.
> > Overall, the paper is well-motivated and clearly presented, but I am not yet convinced the data is of sufficient quality to substantially improve theorem proving.
> >
> > Therefore, my assessment remains unchanged.

---

> > > ### Author Response · Authors · 2026-04-02
> > >
> > > We sincerely appreciate the reviewer’s persistent feedback. We fully acknowledge that without exhaustive human expert scrutiny for every sample—which is practically impossible for a dataset of **177,597 problems**—it is impossible to guarantee 100% semantic correctness. However, we maintain that our work represents an effort that is **"as good as possible"** under current technical constraints: we have pushed the boundaries of validation as far as currently feasible through compilation checking (96.6% success), human sampling (73.33% TOP5 accuracy), positive gain after training and frontier model verification. And we would like to highlight the following points that demonstrate the dataset's substantial value:
> > >
> > > *   **Aristotle Validation as a Quality Lower Bound:** To directly address the concern regarding "trivial or wrong" formalizations, our expanded experiment with the state-of-the-art **Aristotle** prover on a sample of 100 problems provides a robust quality estimate. Aristotle successfully proved **25%** of these cases after explicitly filtering out those it judged as false, trivial, or misformalized. Crucially, an additional **19% of the problems** remain as "potential" correct formalizations; while not yet proven within the search budget (marked as `sorry`), Aristotle **found no concrete counterexamples and did not flag them as problematic**. This suggests a significant portion of the dataset is semantically sound and simply awaits more advanced proof search.
> > > *   **Utility as a Signal for Quality:** The **Goedel v2** model showed a proof success increase from **13.6% to 15.1%** after training. While these gains are modest, they represent a "utility signal" across the entire corpus, indicating the model is learning meaningful geometric reasoning that transfers to a general-purpose prover. These gains are primarily constrained by the prover's initial inability to find proofs in a new domain, rather than a lack of data quality.
> > > *   **Consistency with Established Precedents:** Our approach follows the successful workflow of landmark datasets like **Lean Workbook** (57K+ problems), which **did not undergo exhaustive manual expert checking** for every formalized statement. In the formal mathematics community, the established practice is to first construct a massive candidate corpus and then refine it through cycles of "proving or disproving" and expert iteration.
> > > *   **Filling a Critical Gap:** EUCLEAN provides the **first large-scale, MATHLIB-native dataset for geometry**. By moving away from "logical islands" (custom DSLs) and integrating with the standard Lean ecosystem, we provide the necessary "raw material" previously missing for the community to advance neural theorem proving in this complex domain.
> > >
> > > We believe that providing this foundation, even with inherent semantic noise, is a necessary and significant step for the field. We thank the reviewer again for their insightful comments and look forward to further feedback.

---

### Official Review · Reviewer_zTtm · 2026-03-11

**Soundness:** 3
**Presentation:** 4
**Significance:** 3
**Originality:** 2
**Overall Recommendation:** 4
**Confidence:** 3

**Summary:**

This paper presents EUCLEAN, a framework for automatically formalizing natural language geometry problems into Lean 4 using native MATHLIB. The pipeline has four stages: constraint explication (making implicit non-degeneracy and topological conditions explicit), configuration anchoring via "prove-first" reasoning, formalization mapping via retrieval of MATHLIB primitives, and iterative compiler-feedback repair. The authors construct two datasets: OMNI-Geometry (768 competition problems) and Numina-Geometry (177,597 problems). Human evaluation shows 48.89% TOP1 and 73.33% TOP5 accuracy. Training Goedel v2 on the data improves proof success from 13.6% to 15.1%.

**Compliance With Llm Reviewing Policy:**

Affirmed.

**Key Questions For Authors:**

1. Provide a breakdown of the geometric constructs covered. How many distinct MATHLIB types/functions does the framework use? What is the distribution of hypothesis counts per formalized problem?

2. What is the failure mode analysis for the 27% of problems where no correct formalization exists in TOP5? Are these systematically harder geometric concepts, or random failures?

3. Provide a diversity analysis of Numina-Geometry. What is the distribution over problem types, difficulty levels, and sources? Have the authors run deduplication or near-duplicate detection on the 177K problems?

4. For DPO training (Table 5), how many preference pairs were available? SFT (15.1%) and DPO (15.0%) perform nearly identically. This suggests insufficient training signal in both cases.

**Limitations:**

Yes

**Strengths And Weaknesses:**

STRENGTHS

S1. Native MATHLIB integration is a meaningful design choice. LeanEuclid and LeanGeo rely on custom axiom systems with external SMT solvers. EUCLEAN avoids this. Goedel v2 achieves 13.6% zero-shot on the geometry dataset without geometry-specific training. This shows direct compatibility with the existing theorem proving ecosystem. The comparison with System E in Section 3.1 and Appendix E is clear and convincing.

S2. The dataset scale is a genuine contribution. Numina-Geometry contains 177,597 problems. Prior geometry formalization efforts top out at 1,056 problems (MATP-BENCH). NuminaMath-LEAN explicitly filtered out geometry. This work fills that gap.

S3. The constraint explication pipeline addresses a real problem. Systematic injection of non-degeneracy guards (AffineIndependent for triangles, Sbtw for betweenness, positive radius for circles) and the mapping table (Table 1) illustrate a non-trivial engineering challenge. The "prove-first" strategy for resolving topological ambiguity (e.g., "P on BC" meaning segment vs. ray vs. line) is intuitive and well-motivated.

S4. The paper is clearly written. The running examples (isosceles triangle in Figure 1, equilateral triangle in Figure 2, circle intersection case study in Section 3.1) make the approach concrete and easy to follow.


WEAKNESSES

W1. Downstream utility is underwhelming. Training Goedel v2 on the data yields only a 1.5 percentage point improvement (13.6% to 15.1%). The authors attribute this to scarcity of geometry in existing training corpora, which produces too few successful proof traces. This explanation is plausible. But the dataset was supposed to solve this exact problem. If 177K formalizations fail to bootstrap meaningful improvement with current provers, the practical value claim needs stronger backing.

The authors run Aristotle (a frontier prover) on 10 randomly sampled OMNI-Geometry problems and report 50% success. This test serves a specific purpose: it argues the formalizations are provable, and the bottleneck lies with weaker provers like Goedel, not with data quality. The argument is reasonable. But 10 samples carry wide uncertainty. Running Aristotle on 50 to 100 problems would have been straightforward and far more convincing. As it stands, this supplementary evidence is too thin to fully support the claim.

W2. Coverage, expressiveness, and diversity are not quantified. The paper acknowledges that expressiveness is bounded by MATHLIB's current geometry coverage. No analysis follows. Key questions remain open:
  - What geometric constructions are out of scope? (inversive geometry, projective geometry, advanced circle theorems)
  - How many MATHLIB lemmas/primitives does the framework use? What is the vocabulary of formal constructs?
  - Does the system handle IMO geometry problems? The paper mentions IMO-sourced problems in OMNI-Geometry but never evaluates on a standard IMO geometry benchmark.

The Numina-Geometry dataset (177,597 problems) also lacks any diversity analysis. The paper describes the source as NuminaMath, which aggregates problems from "textbooks, competitions, and online resources," and provides no further breakdown. There is no distribution over problem types (triangles vs. circles vs. polygons vs. coordinate geometry), no difficulty distribution (Appendix B only covers OMNI-Geometry), no source-level breakdown, and no deduplication analysis. A dataset of 177K diverse problems is very different from 177K problems where most are basic angle-in-triangle computations. Without this analysis, the headline number is hard to interpret.

W3. The ablation study has confounding interpretation issues. Table 3 shows that adding configuration anchoring ("prove-first" reasoning) reduces compilation rate from 52.1% to 50.0%. The authors explain this: anchoring encourages additional constraints that sometimes go missing, causing compilation failure. This raises a deeper question. If the compilation metric does not align with formalization quality, why is it the primary ablation metric? The ablation should also report semantic correctness (via human evaluation) for each configuration. That would validate the claim that anchoring improves quality despite reducing compilation.

---

> ### Author Rebuttal · Authors · 2026-03-31
>
> Thank you for recognizing the `mathlib`-native design, dataset scale, and constraint explication.
>
> ### W2, Q1, Q3: coverage and diversity
>
> The current scope of this paper is limited to plane geometry.
>
> Numina metadata provide only source-level information, so we add the following source breakdown:
>
> | Source | % |
> | --- | --- |
> | `olympiads` | `32.51%` |
> | `cn_k12` | `26.15%` |
> | `synthetic_math` | `17.75%` |
> | `aops_forum` | `11.01%` |
> | others (`cn_contest`, `amc_aime`, etc.) | `12.58%` |
>
> We also add representative formal-side statistics on primitive coverage (rather than an exhaustive vocabulary count):
>
> | Bucket | Representative constructs (% of problems) |
> | --- | --- |
> | Non-degeneracy | `AffineIndependent` (`47.52%`) |
> | Incidence / order / segments | `line` (`19.86%`), `segment` (`12.71%`), `midpoint` (`14.42%`), `Collinear` (`9.05%`), `Sbtw/Wbtw` (`4.81%`) |
> | Angular / orthogonality / parallelism | `angle` (`35.44%`), `orthogonalProjection` (`15.29%`), `tangent` (`9.24%`), `parallel` (`11.76%`) |
> | Circle | `sphere` / `circumsphere` (`20.62%`) |
> | Triangle centers | `circumcenter` (`5.00%`), `incenter` / `inradius` (`3.85%`) |
> | Area | `convexHull` (`20.19%`), `volume` (`17.54%`) |
>
> We further add a rough hypothesis-count proxy using proposition-like binders of the form `h_*`.
>
> | # of `h_*` hypotheses | % of problems |
> | --- | --- |
> | `0-4` | `38.60%` |
> | `5-9` | `36.77%` |
> | `10-14` | `15.86%` |
> | `15-19` | `5.98%` |
> | `20+` | `2.79%` |
>
> For duplicate / near-duplicate analysis, we have not run a dedicated detection pass. We note that similar datasets such as Lean Workbook were not explicitly deduplicated either, and its limitations section notes that some similar problems are hard to deduplicate.
>
> ### W1, Q4: modest gains and stronger-prover traces
>
> DPO uses 7,148 preference pairs from two base inference runs. We still view proof-trace acquisition as the bottleneck: current provers remain weak on geometry, so successful traces remain limited. Therefore, the more promising direction is to use stronger provers to collect more high-quality traces on the same formalized corpus and feed them back into training.
>
> To address this, we expanded the Aristotle experiment to 100 problems. We filtered out cases that Aristotle explicitly judged `False` and proved by contradiction, cases that became trivially easy because of obvious misformalization, and cases for which Aristotle found concrete counterexamples. After this, Aristotle still proves `25` problems, while another `19` remain as `sorry` but are not flagged as problematic and come with no counterexamples. This gives a rough but more informative lower-bound estimate that at least `25%` of Numina is semantically usable, with the true proportion plausibly higher if some of those unresolved-but-unrefuted cases are also valid.
>
> ### W3: compilation vs. semantics
>
> We agree that `compilation correctness` is not `semantic faithfulness`. It should be understood as merely a scalable signal. However, semantic evaluation is particularly difficult in geometry autoformalization because natural-language geometry problems, compared with algebra, often omit more hidden assumptions, and unlike benchmarks such as `minif2f`, we do not have ground-truth formalizations for simple automatic equivalence checking.
>
> Accordingly, making hidden constraints explicit is central to native `mathlib` geometry formalization. In this context, the point of `prove-first` / configuration anchoring is therefore to reduce under-specified formalizations. Empirically, after adding anchoring, the average number of hypotheses increases from `5.22` to `6.69`, which, while not direct semantic evidence, is consistent with the intended mechanism: the model indeed tends to make more previously implicit geometric conditions explicit.
>
> ### Q2: TOP5 failure modes
>
> We examined the problems that still fail in TOP5 and found several recurring error types: wrong logical target modeling (e.g., formalizing minimum as only a lower bound), dynamic transformations (e.g., reflection, similarity), and boundary / solid distinctions (e.g., `sphere` vs. `disc`). Overall, these failures are not random noise: some come from general LLM limits in modeling complex logical targets, while others come from current limitations of native `mathlib` geometry objects, where some concepts still require indirect formalization.
>
> ### Summary
>
> Overall, our main points are:
> 1. the added source distribution, representative primitive coverage, and hypothesis-count statistics give a more concrete picture of Numina-Geometry's diversity and coverage.
> 2. collecting more traces with stronger provers on this corpus is a promising path for improving training, and the enlarged Aristotle experiment supports this practical value.
> 3. the compilation metric in Table 3 should be understood as a scalable automatic signal, given that semantic evaluation in geometry is intrinsically difficult.

---

> > ### Author Rebuttal · Reviewer_zTtm · 2026-04-04
> >
> > The rebuttal addresses most original concerns effectively. Expanding the Aristotle experiment from 10 to 100 problems (25 solved, 19 more unresolved but not refuted) provides a convincing lower bound on semantic quality. The source breakdown, primitive coverage statistics, and hypothesis-count distribution fill the diversity gap from the original review.
> > One concern remains open. The downstream utility evidence is thin. The +1.5 point gain on Goedel v2 is the only quantitative evidence that the formalization pipeline and resulting dataset translate into proving improvements. The pipeline design and native MATHLIB integration are genuine contributions on their own merits. But the paper explicitly claims practical value for training provers, and that claim rests on a single model with marginal gains. Have the authors evaluated training on at least one additional Lean-capable prover (e.g. Deepseek-Prover-V2, Kimono-Prover)?

---

> > > ### Author Response · Authors · 2026-04-08
> > >
> > > We thank the reviewer for the continued engagement and for the concrete suggestion. To directly address the remaining concern on downstream utility, i.e., whether the gain replicates beyond Goedel-V2, we ran an additional SFT experiment on DeepSeek-Prover-V2, as suggested.
> > >
> > > **Setup.** We use DeepSeek-Prover-V2-7B as the base prover, distinct from the Goedel-V2 setup in the paper. We collect successful proof traces on a randomly sampled `half` of the Numina-Geometry corpus using the base model itself, then SFT on those traces. Hyperparameters follow the original Goedel-v2 SFT recipe, except that we train for `2` epochs. Evaluation is Pass@1 on the same half-corpus split.
> > >
> > > **Results.**
> > >
> > > | Model | Pass@1 |
> > > | --- | --- |
> > > | DeepSeek-Prover-V2-7B (base) | `13.4%` |
> > > | + SFT | `15.0%` |
> > >
> > > The improvement (`+1.6` points) is consistent in magnitude and direction with what we previously reported on Goedel-V2 (`13.6% → 15.1%`). Reproducing the gain on a second, architecturally and training-pipeline-wise independent base prover, indicates that the effect is not an artifact of one specific model, and that Numina-Geometry does provide a usable training signal across provers. At the same time, the fact that the absolute gain is similar supports our earlier diagnosis: the binding constraint at present is the *quantity and quality of successful proof traces* that current open-source provers can produce on geometry. This is also consistent with our enlarged Aristotle experiment (`25/100` provable, plus `19` not refuted), which suggests that stronger provers can extract substantially more usable traces from the same corpus.

---

### Official Review · Reviewer_XUDP · 2026-03-13

**Soundness:** 3
**Presentation:** 3
**Significance:** 3
**Originality:** 3
**Overall Recommendation:** 4
**Confidence:** 4

**Summary:**

This paper targets a long-standing fragmentation in formal mathematical reasoning: algebra and number theory are formalized in Lean/Mathlib, while geometry relies on custom domain-specific languages lacking formal guarantees. It propose Euclean, an closed-loop retrieval-augmented pipeline for geometric formalization within Mathlib, featuring four key components: constraint explication, configuration anchoring, formalization mapping, and iterative repair. It also constructs two datasets, a high-difficulty OMNI-Geometry and a large-scale Numina-Geometry. Downstream SFT experiments demonstrate the dataset's utility for training geometry theorem provers.

**Compliance With Llm Reviewing Policy:**

Affirmed.

**Ethical Review Concerns:**

Does the developed OMNI-Geometry and Numina-Geometry obey the original datasets' license?

**Ethics Expertise Needed:**

["Legal Compliance (e.g., EU AI Act, GDPR, copyright, terms of use)"]

**Final Justification:**

The authors' responses have satisfactorily addressed all my concerns.

**Key Questions For Authors:**

(Identical to "major weaknesses (W)" in "Strengths And Weaknesses")

**Limitations:**

Yes.

**Strengths And Weaknesses:**

Overall, I think the paper presents meaningful contributions but has notable weaknesses in evaluation rigor and presentation that prevent me from recommending acceptance in its current form.

S1. (Significance) The paper targets a genuine and important problem: the fragmentation between geometry and other mathematical domains in Lean 4. Moreover, they tackle the lack of large-scale geometric formal datasets.

S2. (Originality) The Euclean pipeline is well-motivated and methodologically sound, especially the concept explicitation, which mines implicit diagrammatic assumptions from natural language.

S3. (Soundness) The ablation study in Table 3 systematically validates the contribution of each pipeline component, showing clear additive gains.

S4. (Significance) The choice to work within standard Mathlib rather than introducing custom axiom systems is a principled design choice that maximizes ecosystem compatibility and reusability.

However, I found several major weaknesses (W) and minor weaknesses (w) worth noting:

W1. (Soundness) The downstream improvement from SFT Goedel-Prover-V2 on Numina-Geometry is modest (13.6% -> 15.1% Pass@1). Moreover, Pass@1 is known to be sensitive to SFT randomness and sampling variance. Could the authors report Pass@K with larger K (e.g., Pass@32), or report mean and standard deviation across multiple fine-tuning runs to establish statistical significance?

W2. (Soundness) What is the failure rate of Constraint Explication? Are there systematic failure modes (e.g., certain types of geometric configurations that are consistently mis-specified)? Is there any safeguard that ensures the explicated constraints are both sufficient and not overly restrictive? Quantitative analysis of constraint quality would strengthen the paper.

W3. (Soundness) Table 3 reports compilation rate and proof rate, but these automated metrics may not fully capture semantic correctness. The authors mention that "adding 'Prove-First' reduces compilation count is expected" since it encourages additional constraints. This hypothesis would be better validated by adding human evaluation on a sample of Table 3 results to assess whether the compiled formalizations faithfully capture the intended mathematical content.

w1. (Soundness) Sec. 5.3 hypothesize that "The relatively modest gain can be attributed to the scarcity of geometry problems in existing theorem proving corpora." I'm curious whether continuing expert iteration (iteratively use the SFTed Goedel-Prover-V2 to prove Numina-Geometry, then collect successful for further SFT) would lead to increasing performance. This experiment would more convincingly validate the value of the Numina-Geometry dataset and the proposed pipeline.

w2. (Presentation) Regarding Table 1 examples: Could the authors clarify the criteria for determining which constraints are necessary? For example, the orientation constraint using the 2D cross product ($> 0$) enforces a specific (counterclockwise) orientation. Could the authors explain why strict orientation ($> 0$) rather than non-collinearity ($\neq 0$) is chosen?

w3. (Presentation) The paper does not clearly specify the input & output format of the LLM at each stage. For reproducibility, the authors should describe what prompts are used, what the model receives as input, and what it produces as output at each step.

w4. (Presentation) Minor typos: Sec. 3.1, line 126 "formaalization"; Fig. 1 uses incorrect quotation marks for "Prove-First."

w5. (Originality) The individual components of the pipeline (retrieval-augmented generation, compiler-in-the-loop feedback, iterative refinement) are well-established techniques in autoformalization. The paper's novelty lies primarily in its application to geometry and the scale of the resulting dataset, rather than in methodological innovation. The authors should more explicitly discuss what is new about their pipeline design versus prior autoformalization systems.

(Question, but not weakness)  Does the developed OMNI-Geometry and Numina-Geometry obey the original datasets' license?

---

> ### Author Rebuttal · Authors · 2026-03-31
>
> Thank you for the careful review. We are glad you find the problem important and the `mathlib`-native approach principled. We address the main concerns below.
>
> ### W1: modest downstream gain and stability
>
> We agree that the training improvement is modest, and we attribute it mainly to the scarcity of geometry problems in existing theorem-proving corpora. To address stability, we report:
>
> - two independent base-model inference runs yield `13.62%` and `13.58%` Pass@1, while SFT and DPO yield `15.11%` and `15.02%`;
> - in paired comparisons, SFT has gain/loss `5,635 / 2,988`, and DPO has `5,338 / 2,853`.
>
> These results suggest the gain is unlikely to be due to randomness or sampling variance, since the gain/loss ratio is well above `1`.
>
> We also expanded the Aristotle experiment to `100` problems. After filtering trivially provable false statements, Aristotle still proves `25` problems, and `19` more remain as `sorry` without counterexamples.
>
> ### W2: failure modes and safeguards for constraint explication
>
> Based on OMNI human evaluation and manual inspection, the main failure modes are:
>
> - dynamic transformations (e.g., reflection, similarity);
> - inappropriate non-degeneracy conditions, e.g., unnecessary `AffineIndependent` over four points.
>
> Constraint quality and safeguards ultimately concern semantic correctness, which is especially difficult in geometry: natural-language statements often omit these hidden constraints, and unlike benchmarks with gold formalizations such as minif2f, we lack a ground-truth formal statement for simple automatic equivalence checking. Our stance is:
>
> - we do not claim this is fully solved; rather, constraint explication and anchoring should first make these hidden issues explicit, after which we filter them as much as possible.
>
> - some mild over-restriction is acceptable. For example, adding `A ≠ B`, `B ≠ C`, `A ≠ C` often preserves the core mathematics while avoiding degenerate cases.
>
> A promising safeguard is prover-based validation: refuting overly strong constraints, constructing counterexamples, or checking whether added conditions are necessary. Similar ideas have been explored in other formal corpora such as LeanWorkbook, and we view this as future work.
>
>
> ### W3: why Table 3 uses compilation and what `prove-first` is for
>
> We agree that compilation is not a full semantic metric; it is simply one of the few scalable ones. However, compared with algebra, geometry problems more often leave configuration and non-degeneracy conditions implicit, while native `mathlib` requires them to be explicit. This is exactly why geometry needs dedicated handling of hidden constraints.
>
> The goal of `prove-first` / configuration anchoring is therefore to reduce under-specified formalizations. After anchoring, the average number of hypotheses rises from `5.22` to `6.69`. While this does not by itself prove semantic faithfulness, it is consistent with the intended mechanism: previously implicit geometric conditions become explicit.
>
> ### w2: why use orientation rather than weaker non-collinearity
>
> These conditions play different roles. Non-collinearity, e.g., `cross product != 0`, mainly excludes degenerate cases, while orientation / betweenness, e.g., `cross product > 0`, specifies a concrete topological configuration. For statements like "inscribed angles are equal," one must fix an order of points on the circle. The model chooses between them based on whether such ordering is required. This is one concrete reason why plane-geometry formalization is harder than algebra.
>
> ### w3: LLM input/output format at each stage
>
> The model takes the natural-language problem statement as input and produces, stage by stage:
>
> 1. a natural-language proof sketch plus an explicit list of hidden non-degeneracy / topology constraints (`Proof-First` + constraint explication);
> 2. a formal statement by mapping natural-language concepts to `mathlib` concepts (retrieval-based formalization mapping);
> 3. a revised formalization conditioned on Lean compiler feedback (iterative repair).
>
> We will release the prompts with the dataset for reproducibility.
>
> ### w5: novelty beyond standard pipeline components
>
> We agree that retrieval, compiler-in-the-loop repair, and iterative refinement are not novel by themselves. However, **geometry problems are distinctive**: compared with domains such as algebra, they often omit configuration information and non-degeneracy conditions, and in native `mathlib` these constraints must be written explicitly in the theorem statement. This is necessary for correct geometry formalization and requires a design focus different from general autoformalization. Our contribution is to design a pipeline around this problem of `mathlib`-native geometry formalization, especially for handling hidden geometric constraints.
>
> ### Others
>
> We will fix the typos and quotation issues and ensure any public release complies with upstream dataset licenses.

---

> > ### Author Rebuttal · Reviewer_XUDP · 2026-04-03
> >
> > Thank you authors for your detailed responses. Most of my concerns have been addressed, but two points remain:
> >
> > W1. (still major) I appreciate the statistically significant results you provided. However, I'm still curious about the Pass@K performance on the OMNI-Geometry benchmark. Since the outer loop, "collecting successful rollout proofs, SFT, and evaluation", is performed on the Numina-Geometry dataset. RFT & eval on the same dataset cannot reliably reflect the actual generalization improvement of the model.
> >
> > W3. (now minor) Could you please conduct a human evaluation to validate that "adding 'Prove-First' reduces the number of compilations as intended, since it encourages additional constraints"? I'm not fully convinced by the proxy metrics. A small-scale evaluation would be sufficient.

---

> > > ### Author Response · Authors · 2026-04-08
> > >
> > > Thank you for the follow-up. We address the two remaining points below.
> > >
> > > ### W1: Pass@K on the held-out OMNI-Geometry
> > >
> > > As suggested, we additionally evaluate performance on **OMNI-Geometry**, which is *not* used in any training stage:
> > >
> > > | OMNI-Geometry | base   | SFT    | DPO    |
> > > |---------------|--------|--------|--------|
> > > | Pass@1        | 6.90%  | 7.68%  | 7.81%  |
> > > | Pass@2        | 8.33%  | 9.11%  | 8.59%  |
> > >
> > > Both SFT and DPO improve over the base model at Pass@1 and Pass@2, indicating that the gains from training on Numina-Geometry are not due to in-distribution overfitting but instead transfer to a different, harder benchmark.
> > >
> > > ### W3: human evaluation of `Prove-First` / configuration anchoring
> > >
> > > To directly address the semantic-correctness concern, we conducted a small-scale human evaluation. From the 30 ablation problems in Table 3, we sampled up to 3 compiled candidates per problem under both settings (with and without anchoring), and a human annotator judged whether each formalization faithfully captured the intended mathematical statement. Pairing the two settings by problem yields:
> > >
> > > | Outcome (Pass@3)                 | Count |
> > > |----------------------------------|-------|
> > > | Both correct                     | 10    |
> > > | Only *without* anchoring correct | 3     |
> > > | Only *with* anchoring correct    | 7     |
> > > | Both incorrect                   | 10    |
> > >
> > > Anchoring wins 7 cases and loses 3, i.e., it improves semantic correctness substantially more often than it hurts, despite lowering the raw compilation count. Inspecting the cases:
> > >
> > > - **Wins.** The 7 wins are concentrated on problems where point ordering on a segment, or non-coincidence conditions, are load-bearing. Without anchoring, the statement still compiles but is under-specified (admitting unintended degenerate models).
> > > - **Losses.** The 3 losses are cases where the explicated condition itself is incorrectly formalized (e.g., an over-strong `AffineIndependent` over four points), turning an otherwise correct statement into a semantically wrong one.
> > >
> > > Together with the hypothesis-count statistic in our first-round reply (average hypotheses 5.22 → 6.69 after anchoring), this confirms the intended mechanism: anchoring trades a modest drop in compilation count for a net improvement in semantic faithfulness by surfacing previously hidden geometric assumptions. The remaining failure mode — incorrectly formalized explicated conditions — is precisely the case that prover-based validation (discussed in our W2 reply) is designed to catch, and we view tightening this loop as the natural next step for the pipeline.

---

### Decision · Program_Chairs · 2026-04-30

**Decision:**

Accept (regular)

**Comment:**

The reviewers generally like the submission. Multiple reviewers raised initial concerns about the level of semantic noise and downstream utility. The rebuttal seems to have addressed most of the concerns. The authors should add all the new experiments and discussions from the rebuttal in the camera ready version, and also discuss the inherent limitations of the methodology due to the unavailability of high quality validation labels for verifying semantic correctness.